# Interaction modulation through arrays of clustered methyl-arginine protein modifications

Jonathan Woodsmith[1,2], Victoria Casado-Medrano[2,3], Nouhad Benlasfer[2], Rebecca L Eccles[4], Saskia Hutten[5], Christian L Heine[1], Verena Thormann[2], Claudia Abou-Ajram[5], Oliver Rocks[6], Dorothee Dormann[5,7], Ulrich Stelzl[1,2]

**Systematic analysis of human arginine methylation identifies two distinct signaling modes; either isolated modifications akin to canonical post-translational modification regulation, or clustered arrays within disordered protein sequence. Hundreds of proteins contain these methyl-arginine arrays and are more prone to accumulate mutations and more tightly expression-regulated than dispersed methylation targets. Arginines within an array in the highly methylated RNA-binding protein synaptotagmin binding cytoplasmic RNA interacting protein (SYNCRIP) were experimentally shown to function in concert, providing a tunable protein interaction interface. Quantitative immunoprecipitation assays defined two distinct cumulative binding mechanisms operating across 18 proximal arginine–glycine (RG) motifs in SYNCRIP. Functional binding to the methyltransferase PRMT1 was promoted by continual arginine stretches, whereas interaction with the methyl-binding protein SMN1 was arginine content–dependent irrespective of linear position within the unstructured region. This study highlights how highly repetitive modifiable amino acid arrays in low structural complexity regions can provide regulatory platforms, with SYNCRIP as an extreme example how arginine methylation leverages these disordered sequences to mediate cellular interactions.**

## Introduction

Protein post-translational modifications (PTMs) are known to regulate a vast array of cellular processes governing all facets of human biology. A general three-tier system of PTM addition or removal enzymes (writers and erasers) and PTM binding proteins (readers) is used in a wide variety of differing flavors to vastly increase the functional complexity of the human (Chen et al, 2011; Khoury et al, 2011). Substrates can be targeted by single or multiple modifications at any given time, leading to alterations in expression, localization, activity, or binding partner profiles (Woodsmith & Stelzl, 2014). The collection of tens of thousands of annotated sites has aided computational systematic analysis into both their evolution and interplay with one another (Beltrao et al, 2012; Minguez et al, 2012; Woodsmith et al, 2013). In particular, recorded protein arginine methylation events increased in recent years, facilitating their systematic study (Bremang et al, 2013; Guo et al, 2014; Sylvestersen et al, 2014; Geoghegan et al, 2015; Larsen et al, 2016).

Although issues remain with robust identification of methylation sites (Hart-Smith et al, 2016), both high-throughput dataset collections and small-scale studies (for an extensive review see Biggar & Li [2015]) highlight that arginine mono- and di-methylation impact a wide range of biological processes. Indeed, a recent large-scale study identified that at least 7% of arginines in the expressed proteome are mono-methylated (Larsen et al, 2016). Comprehensive protein methylation-specific interaction networks (Weimann et al, 2013) and methyltransferase knockout studies in cell culture (Shishkova et al, 2017) are beginning to define a wide array of molecular targets to support genetic studies showing the broad impact of the nine identified arginine methyltransferase enzymes (PRMTs) in vivo. PRMT1 and PRMT5 have been shown to be of critical importance, displaying embryonic lethality on knockout (Pawlak et al, 2000; Tee et al, 2010), with most other PRMTs showing different forms of developmental or cellular defects (reviewed in detail in Blanc & Richard [2017]). Furthermore PRMTs are well documented to be dysregulated in cancer, with over-expression of PRMT1, CARM1 (PRMT4), and PRMT5 observed in several studies (Yang & Bedford, 2013).

On a mechanistic level, the relationship between reported methyl-arginine sites and their cognate reader and writer proteins has been previously studied largely using short synthesized peptides in vitro. For example, PRMT1 and PRMT6 have been shown to prefer, but are not limited to, arginine–glycine motifs (RG/RGG motifs [Thandapani et al, 2013], referred to as RG motifs from here onward),

[1]Institute of Pharmaceutical Sciences and BioTechMed-Graz, University of Graz, Graz, Austria    [2]Max Planck Institute for Molecular Genetics, Berlin, Germany    [3]Department of Pharmacology, University of Pennsylvania School of Medicine, Philadelphia, PA, USA    [4]Department of Experimental Medicine I, Friedrich Alexander University of Erlangen-Nuremberg, Erlangen, Germany    [5]BioMedical Center, Ludwig-Maximilians-University Munich, Planegg-Martinsried, Germany    [6]Max-Delbrück-Center for Molecular Medicine, Berlin, Germany    [7]Munich Cluster for Systems Neurology (SyNergy), Munich, Germany

Correspondence: ulrich.stelzl@uni-graz.at; jonathan.woodsmith@uni-graz.at

whereas CARM1 preferentially targets proline-flanked arginines (Osborne et al, 2007; Kölbel et al, 2009; Gui et al, 2013). The methyl-arginine binding Tudor domain has been annotated across 15 proteins to date, with the isolated Tudor domain in key splicing regulator SMN1 showing a binding preference for methylated RG motif containing peptides. Furthermore, isolated Tudor domains bind peptides with multiple modifications with a higher affinity than those with only a single methyl-arginine (Tripsianes et al, 2011; Liu et al, 2012). Indeed, many proteins have now been defined with multiple arginine methylation sites (Larsen et al, 2016), yet the potential interplay between modifications across full-length sequences remains poorly studied. Furthermore, how any cooperation between modified residues mechanistically mediates specific binding preferences in the context of a writer–substrate–reader relationship in human cells is yet unclear.

PTMs have been shown to cluster within intrinsically disordered regions of proteins, a prevalent feature throughout the proteome (Woodsmith et al, 2013). A select few of these regions have been extensively studied and experimental insight into the regulation of most of these unstructured regions is limited. Indeed, although subsequent bioinformatic studies have improved the ways in which to identify functional PTM clusters through integration of distinct data types (Dewhurst et al, 2015), dissecting them mechanistically has proved a major challenge. In vitro peptide studies have provided insight into biophysical binding properties of short modified sequences, but cannot address the full complexity of the long sequences identified in vivo. As the long intrinsically disordered protein sequences that harbor these regions lay outside of the classical structure-function paradigm, novel approaches to understanding their regulation in a cellular context are required. Furthermore, given the vast array of human proteins that contain modified disordered regions are also implicated in neurodegenerative disorders and cancer, understanding how such large regions of low structural complexity are used as regulatory elements is paramount to a better understanding of human cell biology (Babu, 2016).

Here, we highlight that arginine methylation can be broadly separated into two classes based on clustering prevalence, either isolated or within modification arrays. The existence of two distinct classes in methylated residues is supported by differences in the structural context, mutational signatures, and expression analysis of target proteins. We then experimentally dissected in detail the functional requirement of a highly methylated unstructured region in the heterogeneous nuclear RNP (hnRNP) synaptotagmin binding cytoplasmic RNA interacting protein (SYNCRIP). To achieve a comprehensive overview of the entire disordered region, we took a genetic approach to define the binding preferences of a stretch of 19 arginines in the C-terminal SYNCRIP tail using a panel of 37 full-length mutants in quantitative immunoprecipitation experiments. To define both the unmodified and modified states of the arginine array, we leveraged the methyltransferase PRMT1 and the methyl-binding protein SMN1 as functional readouts for arginine and methyl-arginine, respectively. Remarkably, the exact same protein sequence can mediate distinct cumulative binding mechanisms in the modified and unmodified states. Although both interactors increased binding concomitantly with arginine content, unmodified arginines are preferred in continual stretches in direct contrast to

their modified counterparts that function in concert irrespective of their position within the structurally disordered array.

This study reveals how extensive RG repeats within low structural complexity regions can generate cumulative binding mechanisms and, furthermore, how extensive PTMs allow for a second, distinct recognition mode in a single repeat region.

## Results

### Systematic characterization of methyl-arginine array containing proteins

To investigate systematic trends of protein methylation, we initially obtained a list of all arginine and lysine methylation sites available through PhosphoSitePlus (downloaded from PhosphoSitePlus.org June 2017). These PTMs were then mapped to unique Refseq identifiers to give 9339 arginine modifications and 4555 lysine modifications (Table S1). We and others have previously shown that PTMs can cluster across linear protein sequences (Beltrao et al, 2012; Woodsmith et al, 2013), a finding that has been extended to 3D protein structures (Dewhurst et al, 2015). Although protein structures provide a more detailed viewpoint from which to study PTM distributions, they are inherently biased against unstructured regions and limited in number, and as such would impose a large constraint on the PTM dataset. We therefore performed a sliding window analysis that counted the number of modified residues in stretches of 20 amino acids across a linear protein sequence (see the Materials and Methods section). The proportion of total lysine methylation that accumulates in short sequence stretches is consistently lower than that of arginine methylation across multiple modification cutoffs (Fig 1A). To systematically characterize these methylated arginine clusters, we initially investigated their sequence context. As approximately 31% of arginine methylation sites from HEK293T cells were recently shown to be contained within RG motifs (Larsen et al, 2016), we analyzed the propensity of this motif within these clustered sites. Although more dispersed arginine methylation sites (1 or 2 methyl-Rs/20–amino acid window) recapitulate this approximate 30% RG motif content, increasing densities of methylation sites correlate with a noted increase in RG motifs, up to 54% for ≥4 methylation sites per window (Fig 1B). These clustered, RG-motif–driven methylation sites also correlate with a large shift toward structurally disordered regions in comparison with isolated methyl-Rs (Fig 1C).

Arginine-methylated proteins have been shown to be involved in multiple facets of RNA processing and binding, for example, proteins containing RNA recognition motif and RNA helicase RNA–binding domains are preferentially modified (Larsen et al, 2016). We therefore examined the prevalence of methyl-arginine clusters across three large-scale RNA-binding protein (RBP) PAR-Clip studies, which have defined the RBP repertoire (Baltz et al, 2012; Castello et al, 2012; Conrad et al, 2016). We classified protein methylation targets based on maximum methyl-R clustering and observed a sharp increase in the fraction of targeted proteins annotated as RBPs with increasing modification density (Fig 1D). This is likely a function of clustered modifications, as proteins

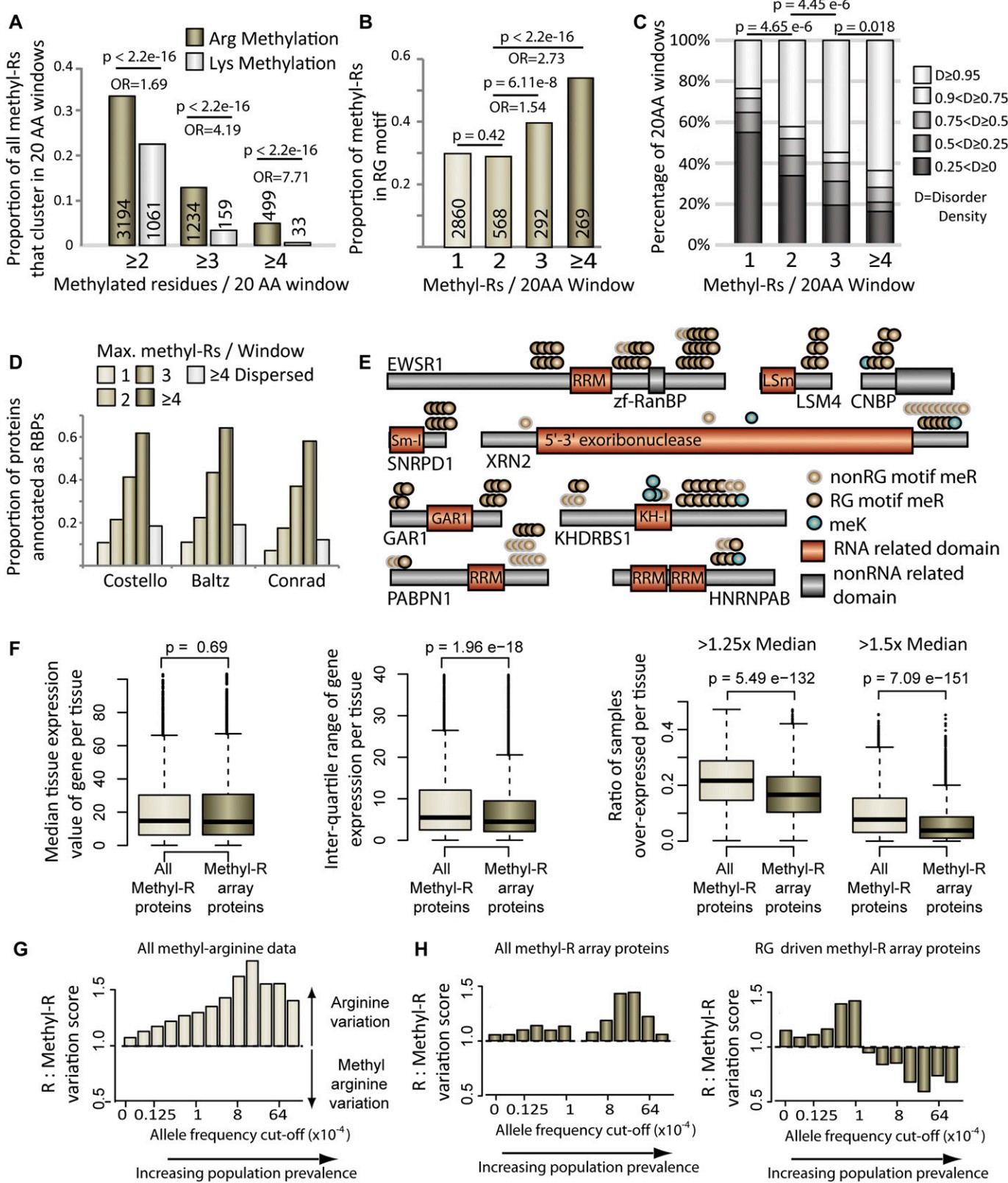

**Figure 1. Methyl-arginine clusters show distinct sequence, expression and mutational signatures.**
**(A)** Bar graph showing the proportional of methyl-arginine or methyl-lysine that clusters within 20–amino acid windows. **(B)** Proportion of methyl-arginines that are contained within an RG motif in each class of 20–amino acid window. P values and odds ratio (OR) for A and B calculated using Fisher's exact test, numbers within the bars

targeted by many, yet dispersed, arginine methylation events have a vastly reduced RBP annotation ratio (Fig 1D).

We next sought to define methyl-R clusters in full length protein sequences. We therefore scanned across each modified protein sequence to further identify proteins that contained multiple or extended arrays over and above a 20–amino acid window (see the Materials and Methods section). Using a cutoff of ≥3 proximal modifications, we systemically defined 313 methyl-R arrays distributed across 273 proteins, containing a total of 1,600 arginine methylation sites (Table S2). These arrays are distributed over a broad size range up to 182 amino acids in length, with 102 proteins having a methyl-R array longer than 20 residues (Fig S1A). Several proteins contain multiple methyl-R arrays, such as the RNA processing proteins EWSR1 and GAR1 (Fig 1E). Although RG motifs are highly prevalent in many arrays, motif analysis of non-RG–driven methylation arrays suggest that CARM1 may also mediate modification clustering (Fig S1B).

We next sought to further characterize these targets of clustered arginine methylation through comparison with their non-clustered methyl-R counterparts. As highlighted above, methyl-R arrays largely appear in regions of low structural complexity away from classical function protein domains (Fig 1C). Proteins containing these disordered regions have been shown to be under tight expression regulation in lower eukaryotes (Gsponer et al, 2008; Vavouri et al, 2009). We therefore used the recent Genotype-Tissue Expression (GTEx) gene expression dataset (Battle et al, 2017) to observe whether this trend holds for human genes targeted by clustered arginine methylation. In the GTEx dataset, each gene is associated with multiple individual samples per tissue, allowing characterization of expression variance across individuals within multiple distinct cellular environments. We first established an analytical framework to control for overall expression patterns of targeted proteins (see the Materials and Methods section). We then characterized the median expression values of clustered methyl-R targets across 51 distinct tissues (clustered methyl-R proteins, Fig 1F, see the Materials and Methods section). For comparison, we sampled the same number of genes from the non-clustered methyl-R target genes, using a randomization protocol that generated a statistically indistinguishable control dataset (all methyl-R proteins, Fig 1F). When comparing the interquartile range of these two groups, we observed overall that proteins containing methyl-R arrays have more tightly controlled gene expression variance (middle panel, Fig 1F). Furthermore, the ratio of samples that show overexpression for a given gene is lower for the methyl-R array containing proteins (cutoff 1.25× and 1.5× median expression value, rightmost panel Fig 1F). This analysis suggests that the two classes of methylation target proteins discovered based on PTM clustering also show a distinguished gene regulatory signature, with proteins harboring methyl-R arrays under more tight expression regulation across tissues.

Finally, we turned to examine patterns of genetic variation on both classes of arginine methylation. Large-scale genome and exome sequencing events have recorded the population prevalence (allele frequency) of millions of genetic variants in healthy individuals (Lek et al, 2016). These allele frequencies can act as proxies for the importance of specific amino acid residues affecting critical protein functions (Woodsmith & Stelzl, 2017); in general, critical residues should seldom be targets of missense mutation in healthy individuals. Here, non-methylated arginines in targeted proteins act as control for amino acid– and gene-specific mutation rates. When comparing the proportion of mutated nonmodified arginines to mutated methyl-arginines across the entire methylation dataset, we observe an increasing ratio indicating a relative increase of mutated nonmodified arginines at increasing population prevalence (left panel, Fig 1G). This is to be expected, as the exact identity of individual post-translationally modified residues will generally be more critical than their nonmodified counterparts. Interestingly, when repeating this analysis for arginines contained within methyl-R arrays, this trend is substantially reduced (left panel, Fig 1H). Furthermore, when we look at those methyl-R arrays driven by RG motifs, the ratio of mutated modified arginines is actually higher than its nonmodified counterpart at higher allele frequencies (right panel, Fig 1H). This analysis indicates that arginine residues targeted by methylation present in arrays are more variable in comparison with the bulk of methylated arginines outside of methyl-R arrays regions. As such, in the context of arginine methylation, the exact amino acid identity at any position where an arginine is present in these arrays is likely less critical than of more isolated methyl-arginines counterparts.

In summary, based on sequence, regulatory and genetic signatures this systematic in silico analysis provides evidence that protein arginine methylation occurs in two classes; methylation events that act in structurally complex regions but in relative isolation to one another, and arrays of arginine methylation where modifications may act in concert to regulate otherwise structurally less defined, low information protein sequences. To understand mechanistically how such extensive stretches of modifications can function in the cell, we sought to experimentally characterize a methyl-R array containing protein.

### Identification of methyl-arginine binding proteins for highly methylated hnRNP SYNCRIP

Using short chemically modified peptides in vitro has shown that clusters of up to four methylated arginine residues distributed across 20 amino acids can markedly increase methyl-binding

---

are the total methylation sites in each bar. **(C)** Bar graph showing the disorder distribution in 20–amino acid windows defined by the number of methyl-arginines they contain. P values calculated using two sided Mann–Whitney–Wilcoxon test. **(D)** Bar graph showing the ratio of each protein class that are annotated as RBPs in each of the three named studies. Dispersed methyl-arginine targets have ≥4 methylation sites but no clustering propensity. **(E)** Schematic diagrams of clustered methylation target proteins. **(F)** Median expression, inter-quartile range and overexpression analysis (left pair: 1.25× median cutoff; right pair: 1.5× median cutoff) of clustered methyl-arginine proteins and an equal-size randomly sampled comparison group from all other methylated proteins. P values calculated using two sided Mann–Whitney–Wilcoxon test. **(G)** Bar chart showing the ratio of arginines that have an annotated missense variant in comparison with the ratio of methyl-arginines that have an annotated missense variant at differing allele frequency cutoffs. **(H)** As in (G) but only using the arginines present in proteins targeted by clustered methyl-arginine windows (all data left, RG driven windows right).

domain interaction affinity (Tripsianes et al, 2011). Yet, it is presently completely unclear whether low structural complexity regions use extensive methyl-R arrays stretching over dozens of amino acids for multiple independent regulatory events or whether they cumulatively combine to increase the regulatory capacity of the entire region. Not only are many of these large methyl-R arrays unamenable to in vitro peptide studies, it is of considerable interest how large low structural complexity regions receive and transmit information in the absence of a defined structure (Babu, 2016). We therefore sought a highly methylated protein to be able to characterize these extensive disordered regions in a larger context.

Previously, we identified candidate methyltransferases for a large panel of target proteins using Y2H-Seq (Weimann et al, 2013). We cross referenced proteins with the highest methylated arginine density with the arginine methyltransferase interaction results and identified the hnRNP SYNCRIP (HNRNPQ), a robust PRMT1 interactor, in the intersection for detailed hypothesis-driven investigation (Fig S1B). SYNCRIP has a total of 18 putative RG methylation target motifs spread across 106 amino acids within its disordered C-terminal tail (Fig 2A and B, plus one R followed by an A). 15 of the arginines have been shown previously to be methylated, spanning the entire length of its C-terminal tail both in vitro and in vivo by PRMT1, seven of which by independent studies (to date eight arginines observed in both the mono- and dimethylated state, six in the mono-methylated state only, and one on in the dimethylated state (Weimann et al, 2013; Hornbeck et al, 2015; Larsen et al, 2016). Five arginines within the array have been shown to be mutated in healthy individuals (Fig 2A, lower panel and Table S3), and SYNCRIP shows very tight expression regulation (Fig 2C), identifying it as a true representation of the bioinformatic trends observed above.

Based on previous structural and biochemical studies of methylated arginines present in RG type repeats, we hypothesized the C-terminal tail of SYNCRIP was required for binding to one or more methyl-binding domains containing proteins (MeBPs). We screened full-length protein-A–tagged SYNCRIP against a panel of 21 luciferase-tagged putative or bona fide MeBPs in a high throughput immuno-precipitation LUMIER-type assay (Hegele et al, 2012), allowing a quantitative readout of multiple protein–protein interactions in a 96-well format (see the Materials and Methods section). The vast majority of MeBPs showed only a low signal in the LUMIER experiment, representing background binding in the assay (Fig 2D). Although two of the four Tudor domain–containing proteins tested showed no interaction signal (green dots Fig 2D, SPF30 [SMNDC1] and PHF19), SMN1 and PHF1 both showed high interaction readout clearly distinct from the background distribution and were verified across repeat assays (Fig 2D). SMN1 has been previously observed to interact with multiple unrelated methylated arginine peptide sequences (Friesen et al, 2001; Tripsianes et al, 2011; Liu et al, 2012) and reported to interact with full-length SYNCRIP (Rossoll et al, 2002), but any methylation dependency of the SYNCRIP–SMN1 interaction is unclear. The PHF1 Tudor domain has been structurally characterized in complex with a histone 3–derived methyl-lysine peptide (Musselman et al, 2012). As SYNCRIP has been reported to be both lysine and arginine methylated (Fig 2A), we tested the methylation dependency of both interactors by mutating residues critical for methyl binding in the β-barrel Tudor structure of each MeBP (TD mutants, Fig S2A and B [Tripsianes et al, 2011; Musselman et al, 2012]). We tested these TD mutants using the LUMIER approach alongside a disease-associated mutant perturbing SMN1 dimerization that is critical for function (DD mutant [Burghes & Beattie, 2009]). Mutations in the β-barrel structure markedly reduced the SYNCRIP interaction signal without affecting the expression of either protein, suggesting that these interactions are methylation dependent (TD mutants, Fig 2E). Although SYNCRIP shows no self-interaction in this assay, robust SMN1 homo-oligomerization is required for a WT SYNCRIP binding signal

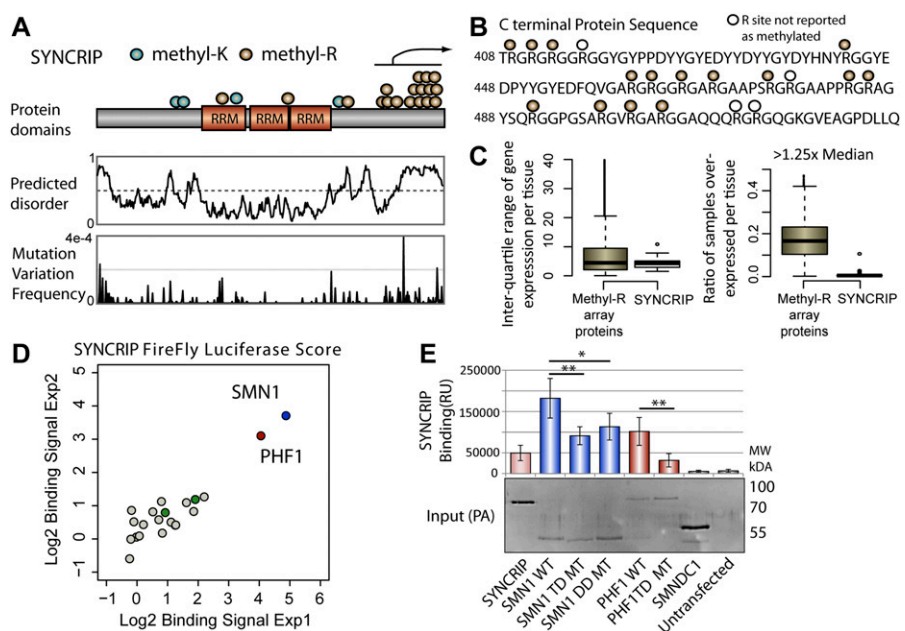

**Figure 2. Identification of MeBPs that interact with SYNCRIP.**
**(A)** Schematic representation of methylation sites annotated on the hnRNP SYNCRIP. Middle panel: predicted protein disorder of SYNCRIP. Lower panel: population max frequencies of genetic variation across SYNCRIP taken from the Gnomad dataset. For ease of representation, the y-axis is truncated to $4 \times 10^{-4}$. **(B)** Amino acid sequence and arginine modifications in the C-terminal tail of SYNCRIP. **(C)** SYNCRIP GTEx expression data: inter-quartile range values (left panel) and overexpression ratios (right panel), for comparison with data obtained for all clustered methyl-R target proteins. **(D)** LUMIER experiments showing SMN1- and PHF1-binding signals separate from the majority distribution of other tested MeBPs. Green dots represent non-binding Tudor domain–containing MeBPs (SMNDC1 and PHF19). **(E)** LUMIER experiment testing SMN1 and PHF1 mutants against WT SYNCRIP. Input Western blot to show expression of protein A tagged constructs. Error bars represent highest and lowest observed values, two sided Mann–Whitney–Wilcoxon test used to determine statistical significance. *$P < 0.05$, **$P < 0.01$. TD, Tudor domain; DD, dimerization domain.

(DD mutant Figs 2E and S2C). This is in line with previous literature suggesting that functional hnRNP particles are disrupted by mutations in the SMN1 oligomerization domain (Burghes & Beattie, 2009). In the SMN1 binding assay the TD mutant also showed a reduced dimerization signal (Fig S2); therefore, we looked for further evidence to support the methylation dependency of the interaction. Furthermore, as this study focused on the function of methyl-R arrays in protein sequences, the likely methyl-lysine–dependent SYNCRIP-PHF1 interaction was not pursued further.

## SYNCRIP arginine methylation function in cell culture

To characterize the function of the SYNCRIP methyl-R array in cells, we created HEK293T cells stably expressing HA-STREP tagged WT and lysine to arginine mutant SYNCRIP. Here, we generated stable cell lines expressing arginine to lysine mutated SYNCRIP with either a small or intermediate number of the original arginines remaining in the C-terminal tail (6 and 14 arginines, respectively. 6 Rs mutant, remaining Rs: R409, R411, R413, R416, R475, R477. 14 Rs mutant, 5 mutants; R443K, R475K, R477K, R511K, R513K). We then immunoprecipitated exogenously expressed WT and mutant SYNCRIP in HEK293T cells to assay its methylation status and endogenous SMN1 binding. Immunoprecipitated WT SYNCRIP showed a strong signal with the pan-methylated-arginine antibody in HEK293T cells, a signal that was markedly reduced by the chemical methylation inhibitor Adox (Fig 3A). In support of the methylation-dependent nature of the interaction, precipitated endogenous SMN1 signal was abolished in the presence of the inhibitor (Fig 3A). This pharmacologically inhibited methylation signal was mimicked by reducing the number of arginines present in the C-terminal tail of SYNCRIP (two rightmost lanes Fig 3A). Methylation was undetectable above background levels on the 6R mutant, yet was partially rescued in

a mutant containing 14 arginines. In agreement with the pull-down of WT SYNCRIP in the presence of Adox, mutant SYNCRIP containing only six arginines in the C-terminal tail showed no SMN1 binding above background levels, whereas the interaction was partially rescued in the mutant containing 14 C-terminal tail arginines. This experiment importantly shows that SYNCRIP can be methylated by endogenous PRMTs under standard (nonstress) conditions and is subsequently bound by endogenous SMN1 in a methylation-dependent manner in mammalian cells.

PRMT1 has been previously shown to bind and methylate SYNCRIP in this C-terminal region in vitro, making it a strong candidate to mediate the methylation observed here. PRMT1 knock-down is toxic to cells and can cause substrate scavenging by other PRMTs, leading to complications in obtaining and interpreting results from standard genetic approaches (Dhar et al, 2013). To ascertain whether PRMT1 produced in live cells is active against SYNCRIP, we purified PRMT1 produced in HEK293T cells for use in an in vitro methylation assay. Bacterially produced, and as such highly likely unmethylated, SYNCRIP was incubated with PRMT1 immunoprecipitated from HEK293T cells using a STREP-HA tag. As can be seen in Fig 3B, PRMT1 could bind to bacterially expressed, unmethylated SYNCRIP independently of exogenous S-adenosyl-l-methionine (SAM), the substrate required for methylation. This suggests that neither the cofactor nor priming methylation events are absolutely required for PRMT1 binding. In the presence of SAM, the methyl-arginine signal greatly increased, indicating SYNCRIP methylation by PRMT1 produced from live cells (Fig 3B).

To characterize the SMN1-SYNCRIP interaction further, we used the split-enhanced yellow fluorescent protein (EYFP) system to assay this binding in live cells. SMN1 and SYNCRIP were tagged with N and C-terminal sections of EYFP that do not individually fluoresce, and co-transfected into HeLa cells. In this system, upon SYNCRIP

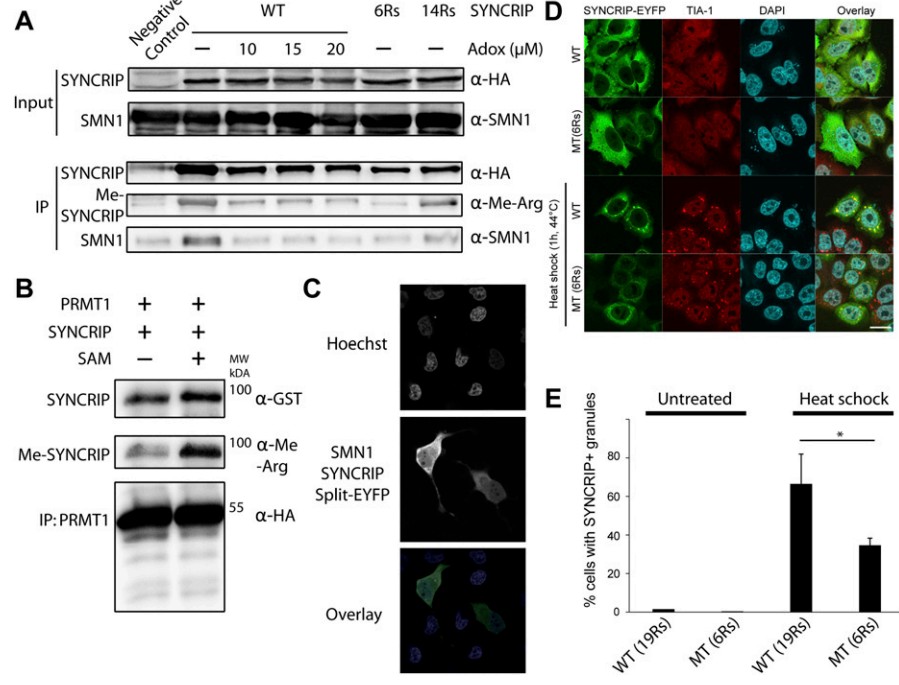

**Figure 3. Characterization of SYNCRIP-SMN1 methylation–dependent interaction.**
**(A)** Methylation dependency of SMN1-SYNCRIP interaction. Immunoprecipitation (IP) of SYNCRIP-HA from HEK293T cells expressing either WT SYNCRIP in the presence or absence of Adox methylation inhibitor, or SYNCRIP containing a reduced number of arginines. **(B)** SYNCRIP binding and subsequent in vitro methylation by HEK293T cell–produced PRMT1 in the presence of cofactor SAM. **(C)** Confocal live micrographs of HeLa cells expressing SYNCRIP and SMN1 each tagged with a part of the EYFP fluorophore, Hoechst used to visualize the nucleus. EYFP reconstitutes upon SYNCRIP-SMN1 binding, allowing subsequent visualization. Scale bar indicates 10 μm. **(D)** Under nonstress conditions, WT SYNCRIP and a mutant containing only six arginines show the same general subcellular localization pattern. Upon heat shock, stress granules (stained with TIA-1 stress granule protein marker) form in both WT and mutant expressing cells. WT SYNCRIP is recruited to stress granules significantly more efficiently than the six arginine SYNCRIP mutant. Scale bar, 20 μm. **(E)** Quantification of wild and mutant SYNCRIP recruitment to stress granules. Number of cells with and without SYNCRIP-positive stress granules was counted (n = 100) in four independent experiments. Graph shows mean of four independent experiments, error bars indicate standard deviation. Two sided Mann–Whitney–Wilcoxon test used to determine statistical significance. *P < 0.05.

and SMN1 binding, the fluorophore fully reconstitutes allowing direct visualization of the interaction's sub-cellular localization using live cell imaging. Here, we observed that SMN1 and SYNCRIP interact both in the cytoplasm and nucleus (Fig 3C), with a slightly stronger fluorescence in the cytoplasm that is in broad agreement with the localization of each protein when expressed alone or in combination (Fig S3). Together, these experiments provide further evidence that SYNCRIP is methylated in HEK293T cells and that this methylation leads to direct binding of SYNCRIP to SMN1.

We next moved to investigate the functional relevance of methylated SYNCRIP in human cells. Multiple hnRNPs and SMN1 have been shown to play roles in stress granule formation in cell culture (Guil et al, 2006; Zou et al, 2011). Furthermore, arginine methylation itself has been implicated in stress granule biology; however, whether it is a driving force for granule formation or more a function of fully formed stress granules remains unclear (Xie & Denman, 2011). We therefore looked to ascertain the subcellular localization of WT and mutant SYNCRIP and endogenous SMN1 under stress and nonstress conditions. We observed stress granule formation under heat shock when expressing both the WT and mutant SYNCRIP constructs using the endogenous TIA-1 stress granule marker protein (Fig 3D). Althuogh WT SYNCRIP was efficiently recruited to stress granules upon heat shock, the SYNCRIP mutant containing only six arginines was only poorly recruited (Fig 3E). Given that SYNCRIP is methylated under nonstress conditions, these experiments suggest that arginine methylation is a prerequisite for efficient hnRNP recruitment to stress granules, not a function of the granule stress response. Although we could observe overexpressed SMN1 recruitment to stress granules, we could not observe endogenous recruitment under several stress conditions (Fig S4).

## Detailed dissection of methylated and unmethylated arginine array in SYNCRIP

Having validated the importance of the SYNCRIP methyl-R array for SMN1 interaction in cells, we sought to systematically dissect the binding mechanisms of the entire disordered region in both its unmodified and modified states in full. To do so, we leveraged the two distinct arms of the arginine methylation regulatory machinery described above, with the methyltransferase PRMT1 and the methyl-binding protein SMN1 acting as functional readouts for the unmodified and modified disordered regions, respectively. Using these two proteins as in-cell molecular probes in the quantitative LUMIER assay would then allow systematic dissection of binding mechanisms of this low structural complexity region.

As the permutations of 19 arginines to lysine mutations is unfeasible to address experimentally ($2^{19}$, >500,000 for position defined permutation), we sought to rationally design mutants based on the cluster proximity of the RG repeats within the array (Fig S5). Using site-directed mutagenesis, we generated a total of 37 mutants in the context of the full-length protein that can be designated into three general subgroups: The first group contains a single, continual stretch of the WT arginine residues, but the number of arginines and the position of the continual stretch varies across the entire tail (top panel, Fig 4A). Conversely, the second group contains a single, continual stretch of arginine to lysine mutants, but the number and position of lysine mutants in the C-terminal tail is varied

(middle panel, Fig 4A). The final smaller group has noncontiguous patches of arginine to lysine mutations distributed across the C-terminal tail (lower panel, Fig 4A).

In agreement with the SYNCRIP mutants used in the endogenous SMN1 immunoprecipitation experiment, the reactivity of a subset of these SYNCRIP mutants with the pan-methyl-arginine antibody correlated well with overall SYNCRIP arginine content. Removing any individual arginine cluster within the array did not abolish methyl-arginine signal, rather the reduction in signal correlated qualitatively with the reduction in methylatable residues (Fig S6). As this SYNCRIP arginine to lysine mutant panel can be methylated in a graded manner under standard conditions, it can act as a good proxy for reduced methylation of full-length SYNCRIP in cultured cells. We therefore screened each full-length mutant for a functional readout of both the unmodified (PRMT1) and modified (SMN1) states of this unstructured region.

Both PRMT1 (Fig 4B) and SMN1 (Fig 4C) LUMIER experiments showed good reproducibility, with mutant expression comparable with WT SYNCRIP and exhibiting low variability (Fig S7). Although SMN1 showed only a weak signal for PRMT1 binding that was comparable with controls, PRMT1 shows a very strong self-interaction signal, in agreement with previous knowledge on its homo-dimerization (Zhang & Cheng, 2003; Thomas et al, 2010; Weimann et al, 2013). Through comparing the binding scores with the mutant sequences, several trends are immediately clear (Heatmaps next to mutant schematic diagrams, Fig 4A). Mutating either N or C-terminal arginines ablated neither SMN1- nor PRMT1-binding completely, only sequentially mutating residues from both terminal groups to leave a small central arginine patch eventually reduced binding to background levels (top panel, Fig 4A). Furthermore, mutation of any individual arginine patch did not reduce the binding signal to background levels with central lysine mutants tolerated in the context of flanking arginines (middle and lower panel, Fig 4A). These experiments suggest a model whereby both the modified and unmodified RG repeat regions mediate their interactions cumulatively, showing an increased binding signal up to restoration of the full 19 WT arginine residues (grey–blue color code in Fig 4B and C).

To systematically test whether both modified and unmodified regions follow this overarching model of cumulative arginine dependency, we then grouped the mutants based solely on the number of arginines that remain in the C-terminus. In good agreement with this model, there is a strong positive correlation between binding score and number of arginines for both SMN1 and PRMT1 (Fig 4D and E). To dissect this further, we then split the SYNCRIP mutants into two sub-categories; one group in which all arginines were present in a contiguous linear sequence, and a second where lysine mutants interrupted the sequence of remaining arginines (noncontiguous). To cover the full spectrum of mutant subgroups, overlapping levels of total arginine were used to further divide each category. Interestingly, although methyl-arginine reader SMN1 shows little difference between the two groups (Fig 4F), the PRMT1-binding signal indicates a clear preference for contiguous arginine stretches, irrespective of the total arginine content (Fig 4G). Importantly, this model refinement still falls within the general cumulative arginine mechanism, as mutants with 15 to 18 arginines present in noncontiguous mutants still show higher PRMT1 binding than mutants with 10–13 contiguous arginine

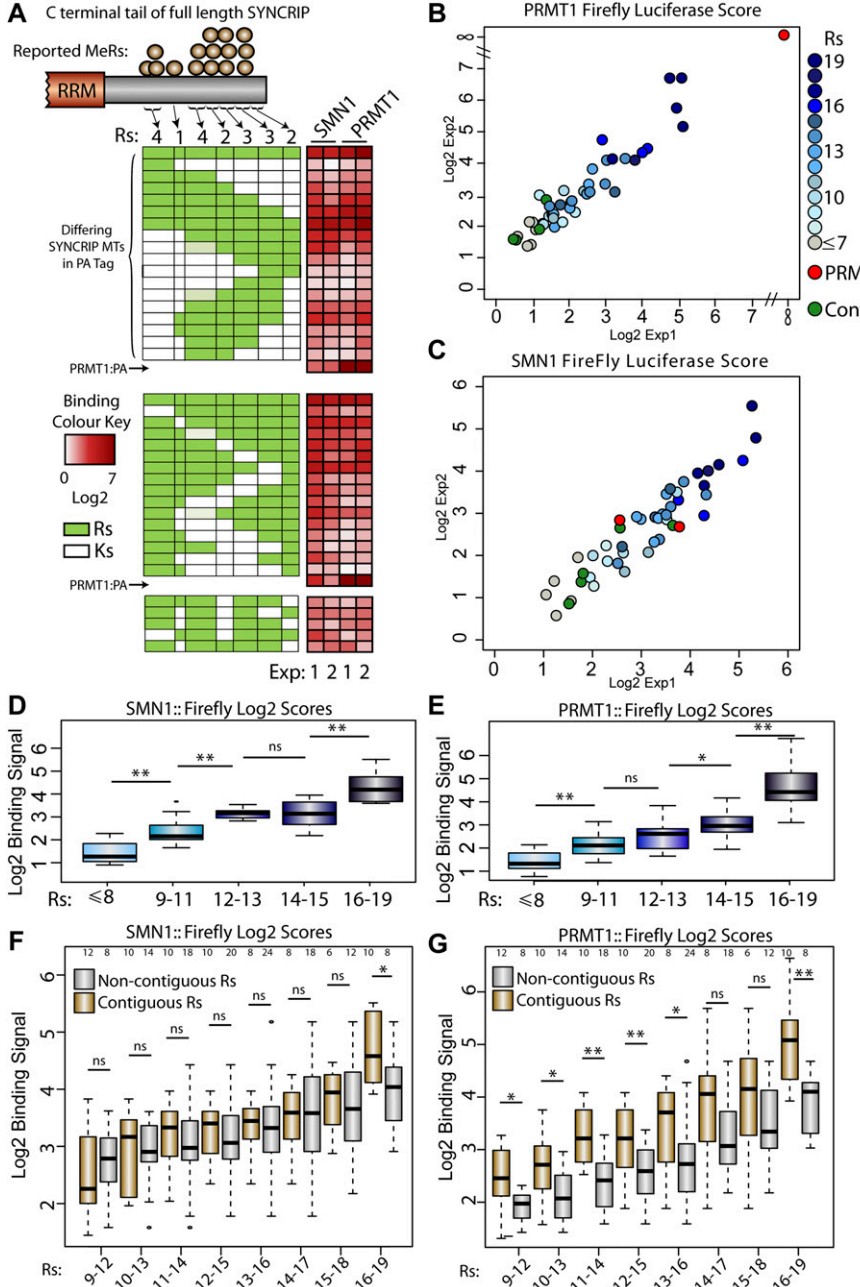

**Figure 4. Systematic dissection of arginine requirement for PRTM1 and SMN1 binding to SYNCRIP.**
**(A)** Schematic diagram representing the R to K full-length mutant constructs and their binding signal in each experiment (white-red heatmap next to schematics). Light green boxes represent R patches where only three of the four arginines were mutated to lysines post-sequence verification. PRMT1 (B) or SMN1 (C) binding signals in medium throughput LUMIER experiment assaying multiple R to K mutations on the binding of full-length SYNCRIP. SMN1- (D) or PRMT1- (E) binding signal box plots for each group of R to K SYNCRIP mutants. Groups designated by the total number of remaining C-terminal arginines. SMN1- (F) or PRMT1- (G) binding signal box plots for subgroups of R to K SYNCRIP mutants. Each subgroup designated by the total number of arginines in either a contiguous or noncontiguous sequence. Numbers at the top of each blot represent the number of data points in each box plot. One sided Mann–Whitney–Wilcoxon test used to determine statistical significance. *P < 0.05, **P < 0.01.

residues (*P* = 0.02, one-Sided Mann–Whitney–Wilcoxon test, Fig 4G). Therefore, through leveraging the arginine methylation machinery as in-cell molecular probes, we can develop an overall model of cumulative methyl binding across more than 100 amino acids of disordered protein sequence, and furthermore, for the first time, differentiate overarching binding preferences of the unmodified and modified RG repeats.

Disordered protein sequences inherently contain little protein structural information, and as such are difficult to experimentally investigate. Here, we used a functional readout for both the methyl independent (PRMT1) and methylated (SMN1) states to show, first, how a large array of RG repeats can be used to generate a cumulative binding capacity within disordered regions and, second, how PTMs can co-opt these same regions using distinct binding preferences to produce a functional output.

## Discussion

Here, we highlight the dual mechanisms arginine methylation uses to regulate protein function. We identified hundreds of candidates annotated with methyl-R arrays and investigated in detail one of the longest methylated arginine stretches identified, distributed across 19 arginines within the disordered C-terminal tail of the hnRNP SYNCRIP.

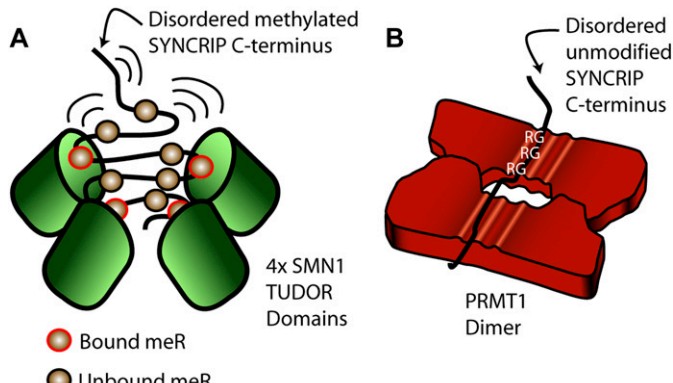

**Figure 5. Model of SMN1 methyl-arginine dependent and PRMT1 arginine dependent SYNCRIP interactions.**
**(A)** Multiple Tudor domains in an SMN1 oligomer independently interact with individual methylated arginine moieties within the flexible SYNCRIP C-terminal tail. **(B)** Multiple sequential arginines in the flexible SYNCRIP tail recognized simultaneously by a PRMT1 dimer.

As much as 40% of eukaryotic proteomes are annotated as disordered protein sequence (Potenza et al, 2015). Proteins harboring such regions have well-established roles in cellular signaling and have been implicated in multiple disease processes (Babu, 2016). However, understanding the function of such long unstructured regions in a cellular environment has proved challenging.

To tackle this problem, we generated a large set of full-length SYNCRIP mutants, allowing investigation into these methyl-R arrays in the context of the full-length protein in a quantitative immunoprecipitation assay. This experimental setup does not provide the detailed biophysical data of in vitro peptide studies; however, the number and length of mutants allows overarching in vivo binding principles to be observed that would otherwise be refractory to experimental investigation. In stark contrast to the canonical single PTM–single function paradigm, no individual modified arginine is absolutely required for an interaction in cultured cells. This trend also holds for the modification-independent RG repeat functional interaction with PRMT1. Furthermore, the interaction signal observed for modified and unmodified disordered sequences increases as arginines are restored up to the 19 present in WT SYNCRIP. This finding suggests that the methyl-R arrays identified proteome wide are functionally driven by a requirement for tunable protein interactions.

We could also further refine this cumulative RG motif requirement to be able to propose distinct models for both the modified and unmodified unstructured arrays that rely on the ability of disordered regions to adopt multiple conformations within the cell. Here, we show that PRMT1, but not SMN1, substantially prefers contiguous runs of RG motifs within an array. The biophysical properties of PRMT1, SMN1, and their RG peptide recognition mechanisms give clues as to the likely origin of these mechanistic differences. Both SMN1 and PRMT1 are known to oligomerise into higher order structures providing multiple binding sites for each (methylated) arginine in each oligomer (Zhang & Cheng, 2003; Burghes & Beattie, 2009; Martin et al, 2012). These oligomers are absolutely required for normal functioning and mutants disrupting the SMN1 basic dimer lead to disease phenotypes

(Burghes & Beattie, 2009). Furthermore, PRMT1 dimerization is strongly interlinked with AdoMet binding and catalytic activity (Thomas et al, 2010; Zhou et al, 2015), therefore detailed biophysical experiments are required to further disentangle dimerization requirements at each step of the methylation reaction. However, these oligomeric structures allow complex binding mechanisms and provide the basis for the general mechanisms proposed here.

A single SMN1 Tudor domain monomer can only accommodate binding to one methylated arginine at any given time, yet the arginine-binding β-barrel domain still shows an increased affinity for a multi-methylated peptide (Tripsianes et al, 2011). Furthermore, as the Tudor domain lacks contacts to residues adjacent to the methylated arginine–glycine mark (Tripsianes et al, 2011), repeated methyl-arginine binding can be independent of local sequence context. Therefore, repeated binding does not necessarily require sequential modification along a protein sequence, only methyl-arginines close in 3D space as present here in the long disordered tail region. In the context of multiple Tudor domains within an SMN1 oligomer, this "one-out one-in" binding mechanism would clearly favor long, multiply modified flexible substrates as each binding pocket could be simultaneously occupied or rapidly rebind dissociated methylated arginines (Fig 5A). In a model with independent recognition events of a single modified residue, multiple binding pockets would aid rapid rebinding of dissociated methylation groups. Furthermore, the interaction would be less sensitive to the linear placement of methylations along a disordered tail that can adopt many conformations, and would consequently mainly be dependent on the total modification level in a confined 3D space. Although single-methylated arginine residues not present in RG motifs have also been shown to recruit SMN1 (Zhao et al, 2016), we hypothesize that many of the long RG arrays identified here will follow similar cumulative arginine-driven binding models.

In contrast to a single methyl-arginine binding to a Tudor monomer, the PRMT1 monomer contains multiple putative RG binding acidic grooves, three of which have been shown to bind a triple-RGG containing peptide with higher affinity than a single-RGG–containing peptide (Zhang & Cheng, 2003). This necessarily constrains the arginines in a physically consecutive peptide as multiple motifs across a linear sequence are simultaneously involved in a binding event (Fig 5B). As such, sequence deviation would likely lead to a lower affinity and reduced catalytic activity, in line with previous observations (Zhang & Cheng, 2003). All of the long SYNCRIP substrates assayed here contain many multi-RG peptides within a single disordered protein sequence, providing multiple opportunities for PRMT1 oligomer recognition. However, a noncontiguous mutant PRMT1 recognition event is more likely to involve disruptive arginine to lysine flanking mutants than a contiguous mutant with the same arginine content, thus providing a less optimal substrate and the lower binding signals observed here. Given the length of the RG-containing arrays identified here, it is plausible that both monomers within a PRMT1 dimer are involved in this recognition event and act in tandem to increase binding strength (Fig 5B).

Long, multiply modified disordered substrates such as the SYNCRIP C-terminal tail fall outside of the classical structure–function paradigm and as such are refractory to direct visualization using standard structural and biochemical approaches. As such, novel approaches are required to untangle exactly how these regions are recognized by the cellular machinery. The genetic approach taken

here provides insight into how such long stretches of methyl-arginine residues function within the cell and further how short peptide binding mechanisms translate into overall interaction preferences in the context of a full-length protein. These models also provide plausible links to the bioinformatic trends we observed for methyl-R array–containing proteins. As protein interactions are driven by protein concentration, tunable interaction mechanisms such as the one described here require tight expression regulation as we observed in the GTEx dataset (Fig 1G). Furthermore, if individual modifications within an array are not absolutely critical for overall function, mutations are less likely to be damaging and subsequently more likely to be observed in population-level genetic data (Fig 1H).

This study represents the most comprehensive dissection of extensive low structural complexity regions present proteome wide to date, and furthermore highlight how the cell uses two distinct binding mechanisms within these disordered sequences to achieve a similar overarching effect; namely a cumulative contribution of each RG repeat to binding strength.

## Experimental procedures

### PTM data collation and analysis
Dataset collation was undertaken as in (Woodsmith et al, 2013). Briefly, data for each PTM was obtained from PhosphoSitePlus (Hornbeck et al, 2015) and integrated with publicly available datasets to obtain a nonredundant list of 13–amino acid sequences (13mers). The central amino acid is annotated as modified in each 13mer and only modified lysine or arginine residues were taken forward to the final analysis (Table S1).

### Iupred disorder analysis
Each RefSeq protein sequence in the analysis was analyzed using the Iupred disorder prediction software (Dosztányi et al, 2005), 0.5 was set as a cutoff to binarise each amino acid into ordered or disordered.

### RBP annotation
RBP annotation has been shown to be variable depending on the experimental set-up. We therefore took three independent experiment studies (Baltz et al, 2012; Castello et al, 2012; Conrad et al, 2016) for the initial analysis. To annotate the RG array–containing proteins as RBPs or not for Fig 1D, we used the list given in (Gerstberger et al, 2014).

### Methyl-R array extraction
>25,000 proteins sequences were computationally scanned in overlapping 20–amino acid windows in the N to C terminus direction. If multiple PTM annotated isoforms were available, the most highly annotated isoform was taken forward. The start of any methyl-R array was defined as any sequence that contained three or more methylated arginines in a 20AA window. The array was then continued unless a 50–amino acid gap between the start of the array and the next methyl-R triplicate appeared. A list of all arrays extracted can be found in Table S2.

### Motif analysis of non-RG–driven methylation arrays
We extracted all non-RG methylation sites from arrays driven by methyl-non-RGs (Fig S1B). We then used icelogo with default settings (Maddelein et al, 2015) to generate a consensus motif, using as background all methylated arginine sites not present in the positive dataset.

### GTEx dataset
We used GTEx_Analysis_v6p_RNA-seq_RNA-SeQCv1.1.8_gene_rpkm.gct for all analyses downloaded from https://gtexportal.org/home/datasets. To aid statistically robustness, we only calculated the median and variance of gene expression for identifiers with >10 samples per tissue, leaving a maximum of 51 tissues per gene identifier. We then used the distribution of the median expression values for the methyl-R array genes as a control for further comparative analysis. We randomly sampled the overall methylation dataset for the same number of genes as present in the methyl-R array target gene set. We then extracted their data from the GTEx dataset, ensuring that for each randomization the distribution of the median expression values across all tissues was statistically indistinguishable from that observed for the methyl-R array genes (example of distribution comparison Fig 1F, left panel). We then compared the distribution of the gene expression variance from the same random sample (Figs 1F and 3, rightmost panels). We repeated the random sampling protocol 100 times, observing the same outcome after each randomization.

### Gnomad dataset analysis
We used gnomad.exomes.r2.0.1.sites.vcf.gz for all analyses downloaded from gnomad.broadinstitute.org/downloads, only taking forward the Gnomad annotated isoform that corresponded to the arginine modified isoform from the PTM dataset collation. As a measure for the likelihood of any mutation occurring at a given arginine, we summed the allele frequencies for all mutations per codon across all identifiers. For any given allele frequency cutoff, we then calculated the ratio of the proportion of mutated non-modified arginines to the proportion of mutated modified arginines. We repeated this analysis for three datasets; all proteins targeted by arginine methylation, all proteins targeted by methyl-R arrays, and all proteins targeted by methyl-R arrays driven by arginines-glycine motifs.

### Cell culture
All cell lines were maintained in a humidified incubator at 37°C with 5% $CO_2$. HEK293T cells were used for all immunoprecipitation experiments. For cellular stress experiments, HeLa cells were used for quantification and HeLa Kyoto cells were used for confocal imaging and were grown in DMEM high glucose GlutaMAX (Invitrogen) supplemented with 10% FCS and 10 $\mu$g/ml gentamicin. For split-EYFP experiments and localization experiments HeLa cells were grown in DMEM containing 10% FCS.

### LUMIER-type experiments
MeBP and SYNCRIP ORFs were transferred to either firefly luciferase-V5 fusion vectors (pcDNA3.1V5-Fire) or protein-A fusion vectors (pcDNA3.1PA-D57), using standard gateway cloning procedures. For co-IP assays, 3 × 10$^4$ HEK293 cells were transiently cotransfected with firefly (75 ng) and protein A (PA; 75 ng) plasmid DNA using Lipofectamine 2000 (Invitrogen) in each well of a 96-well plate. Cells were lysed 36 h after transfection in 100 $\mu$l Hepes buffer

(50 mM Hepes pH 7.4, 150 mM NaCl, 1 mM EDTA, 10% glycerol, 1% Triton X-100 and protease inhibitor [11051600; Roche]) for 30 min at 4°C. Protein complexes were precipitated from 80 $\mu$l cleared cell extract in IgG-coated microtiter plates for 2 h at 4°C and rapidly washed three times with 100 $\mu$l ice-cold PBS. The binding of the firefly-V5–tagged fusion protein (co-IP) to the PA-tagged fusion protein was assessed by measuring the firefly luciferase activity in a luminescence plate reader (Beckmann D TX800, Bright-Glo Luciferase Assay [Promega]). Assays were performed as triplicate transfections. For small-scale LUMIER experiments, the raw output intensities are displayed for each triplicate. For the methyl-binding protein experiment (Fig 3), the background for SYNCRIP-FIREFLY was calculated as an average of the three lowest reported luminescence readings, converted to a Log(2) scale. This was then subtracted from each reported methyl-binding protein value to be able to observe PA clones that reported robust signals above the background distribution. Two PA proteins (CBX1 and BPRF1) were found to be "sticky" in this experimental setup and were subsequently excluded from further analysis (i.e., they showed interactions with a large amount of unrelated proteins, data not shown). For the large mutant SYNCRIP experiments, a triplicate of PA untransfected wells were used to estimate the background Log(2) signal in each plate. This was then subtracted from each mutant output value and the Log(2) signals plotted. Furthermore, we checked the observed interaction distribution could not be explained by a simple linear regression of input against output values (R-squared values SMN1-FIREFLY = 0.0196, PRMT1-FIREFLY = 0.0488). Noninteracting controls used to indicate background binding in the large mutant SYNCRIP experiment were U2AF1, BAT3, and SPATA24.

### Stable cell line generation
SYNCRIP- and PRMT1-tagged constructs were generated in the pcDNA5/FRT/TO/HA STREP vector using standard gateway cloning (Invitrogen) and transfected into HEK293 cells cultured in DMEM + FCS. 48 h after transfection, transformed cells were selected through incubation with 50 $\mu$g/ml hygromycin for 12–20 d. Individual colonies were picked and tested for equivalent protein expression induced with 1 $\mu$g/ml doxycyclin for 24 h, before pooling.

### Endogenous SMN1 immunoprecipitation experiments
For each individual immunoprecipitation, $2.5 \times 10^6$ stable HEK293 cells were seeded in DMEM + FCS (1 $\mu$g/ml doxycyclin). Each dish was then incubated with the required concentration of Adox or DMSO vehicle control for 24–36 h. Cells were then lysed in Hepes buffer (as above for LUMIER-type experiments), and incubated with pre-blocked anti-HA beads (1% BSA, overnight at 4°C) prior to 3× washing in ice cold lysis buffer. Beads were then resuspended in 1.5× sample buffer (18 mM Tris-Cl pH 6.8, 0.6% SDS, 3% glycerol, 1.5% $\beta$-mercaptoethanol, and 0.003% bromophenol blue) before electrophoresis and Western blot analysis.

### In vitro SYNCRIP production
GST-tagged SYNCRIP was expressed in 12.5 ml OverNight Express Autoinduction TB-Medium (+Amp, +CAM) at 37°C (150 rpm) for 20 h. The bacteria culture was then centrifuged at 1,800 $g$ (4°C) to collect the cell pellet. The pellet was then resuspended in 1.85 ml lysis

buffer (50 mM Hepes, 150 mM NaCl, 5% glycerol, 1 mM EDTA, 0.5% Brij 58, 1 mg/ml lysozyme, 2 mM DTT) and incubated on ice for 30 min 350 $\mu$l Bezonase solution (20 mM Hepes, pH 8.0, 2 mM MgCl$_2$, 0.1 U/$\mu$l benzoase) was then added to the lysate before a further 30 min incubation at 4°C and a final centrifugation step at 15,000 $g$ for 30 min at 4°C before the supernatant being stored on ice until further use.

### PRMT1 beads preparation
$6 \times 10^6$ HEK293 cells expressing either PRMT1 were collected, washed once in ice cold PBS, then incubated on ice for 30 min in 0.5 ml lysis buffer (50 mM Hepes pH 8.0, 150 mM NaCl, 10% glycerol, 1% Triton-X 100). The lysate was then centrifuged at 15,000 $g$, 30 min, 4°C before the supernatant being inoculated for 1 h with pre-washed Strep-Tactin beads suspension at 4°C. PRMT1 beads were then washed four times in lysis buffer before being stored on ice until further use.

### SYNCRIP methylation assay
PRMT1 beads were mixed with SYNCRIP bacterial lysate (2:1 by volume) and inoculated shaking (300 rpm) for 2 h at 30°C either in the absence or presence of 20 mM exogenous SAM. The supernatant was removed and stored on ice until further use, remaining beads were resuspended in sample buffer and heated for 5 min at 95°C before storage at –20°C before Western blot analysis. Methylation was detected using anti-mono and dimethyl arginine antibody ([7E6], ab412; Abcam, raised against asymmetrical N$^G$/N$^G$-dimethyl arginine).

### Localization experiments
HeLa cells were transfected with FuGene transfection reagent at 3:1 ratio of DNA:reagent using a standard protocol. Live cell imaging of split-EYFP was undertaken on MatTek dishes 22 h post-transfection with 10 min of Hoescht incubation before visualization. For individual and colocalisation experiments, cells seeded on glass coverslips were fixed 16 h post-transfection with 4% paraformaldehyde. EYFP signal was detected using a chicken anti-GFP antibody (ab13970; Abcam) followed by an anti-chicken Alexa-Flour-488 secondary (Thermo Fisher Scientific). PA signal was detected using rabbit IgG (Santa Cruz) followed by anti-rabbit Alexa-Flour-555 secondary (Thermo Fisher Scientific).

### Confocal microscopy and image processing for cellular localisation experiments
Confocal laser scanning microscopy was performed on a Fluoview 1000 confocal microscope (Olympus) equipped with a UPLSAPO60/1.3 numerical aperture silicon immersion oil immersion lens. Images were taken with the following excitation (Ex) and emission (Em) settings: Hoechst Ex: 405 nm diode laser (50 mW) Em: 425–475 nm; GFP, Alexa-Fluor488 Ex: Multi-Line Argon laser 488 nm (40 mW) Em: 500–545 nm; EYFP, Ex: Multi-Line Argon laser 515 nm (40 mW) Em: 530–545 nm; AlexaFluor555 Ex: 559 nm diode laser (20 mW) Em: 570–625 nm.

### Cellular stress experiments
To avoid formation of stress granules by overexpression, low amounts (10 ng per 24-well) of DNA coding for EYFP-SYNCRIP (WT/6R) was cotransfected with an empty vector (pcDNA3.1-hygro(+); 490 ng per 24-well) using Turbofect (Fermentas) according to the

manufacturer's instructions. Stress treatment was carried out either by heat shock (1 h at 44°C) or by addition of 0.5 mM sodium arsenite (30 min at 37°C). Cells were immediately fixed in 3.7% formaldehyde in PBS for 7–10 min and permeabilized with 0.5% Triton X-100 in PBS. Cells were blocked for 10 min in blocking buffer (1% donkey serum in PBS/0.1% Tween-20) and incubated with primary antibody (monoclonal mouse anti GFP [for detection of YFP-SYNCRIP]: Hybridoma was kindly provided by A. Noegel, Cologne, Germany (Noegel et al, 2004); purified antibody was a gift from M. Kiebler, LMU, Munich. Mouse anti SMN: BD [610646], goat anti ia-1 [G3], Santa Cruz [sc-166247]), and secondary antibodies (Invitrogen/Molecular Probes) diluted in blocking buffer. Washing steps were performed with PBS/0.1% Tween-20. Nuclei were stained with DAPI (0.5 μg/ml; Sigma-Aldrich) and mounted in prolong diamond mounting medium (Invitrogen) before analysis by confocal fluorescence microscopy.

### Confocal microscopy and image processing for cellular stress experiments

Images for colocalization of YFP-tagged SYNCRIP with stress granule markers were acquired by confocal microscopy on an inverted Leica SP8 microscope (Bioimaging core facility of the Biomedical Center), equipped with lasers for 405, 488, 552, and 638 nm excitation. Images were acquired with a 63 × 1.4 oil objective, image pixel size was 59 nm. The following fluorescence settings were used for detection: DAPI: 419–442 nm, Alexa 488/YFP: 498–533 nm, Alexa 555: 598–634, Alexa 647: 650–700 (for quadruple stain) or 649–698 (for triple stain). Recording was sequential to avoid bleed-through using a conventional photomultiplier tube. Confocal images were acquired using LAS X (Leica) and processed using Image J software applying linear enhancement for brightness and contrast. For illustration of the localization of SMN in response to arsenite treatment, a stack of 10 (unstressed) or 12 (+arsenite) sections in 300 nm step size was acquired and projected using the maximum intensity projection function in the LAS X software.

## Supplementary Information

## Acknowledgements

This work was funded by the Max-Planck Society (for U Stelzl), University of Graz (for U Stelzl), Deutsche Forschungsgemeinschaft (German Research Foundation) within the Emmy Noether grant DO 1804/1-1 (for D Dormann), and the Munich Cluster for Systems Neurology (EXC 1010 SyNergy), as well as the Investment Fund and Junior Research Fund of the Ludwig-Maximilians-Universität München (for D Dormann).

### Author Contributions

J Woodsmith: conceptualization, data curation, software, formal analysis, validation, investigation, visualization, methodology, and writing—original draft.
V Casado-Medrano: investigation and methodology.
N Benlasfer: investigation and methodology.
R Eccles: investigation and methodology.
S Hutten: investigation and methodology.
CL Heine: investigation and methodology.
V Thormann: investigation and methodology.
C Abou-Ajram: investigation and methodology.
O Rocks: resources.
D Dormann: resources and methodology.
U Stelzl: conceptualization, supervision, funding acquisition, writing—original draft.

### Conflict of Interest Statement

The authors declare that they have no conflict of interest.

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
