## [Reviewer comments · Life Science Alliance]

Life Science Alliance

Interaction modulation through arrays of clustered methyl-arginine protein modifications

Ulrich Stelzl, Nouhad Benlasfer, Verena Thormann, Claudia Abou-Ajram, Oliver Rocks, Dorothee Dormann, Victoria Casado-Medrano, Christian Heine, Jonathan Woodsmith, Saskia Hutten, and Rebecca Eccles

DOI: 10.26508/lsa.201800178

Corresponding author(s): Ulrich Stelzl, University of Graz and Jonathan Woodsmith, University of Graz

Review Timeline:

Submission Date:	2018-08-27
Editorial Decision:	2018-08-27
Revision Received:	2018-09-05
Editorial Decision:	2018-09-10
Revision Received:	2018-09-11
Accepted:	2018-09-12

Scientific Editor: Andrea Leibfried

Transaction Report:

Please note that the manuscript was previously reviewed at another journal and the reports were taken into account in inviting a revision for publication at *Life Science Alliance* prior to submission to *Life Science Alliance*

Thank you for transferring your manuscript entitled "Cumulative regulatory potential of clustered methyl-arginine protein modifications" to Life Science Alliance. The manuscript was assessed by expert reviewers at another journal before, and the editors transferred those reports to us with your permission.

The reviewers who evaluated your work at the other journal noted that the provided definitive insight at the systems level remains limited. They further provided constructive input on how to strengthen the current dataset. Based on these reports already at hand, we would like to invite you to submit a revised version of your manuscript for publication in Life Science Alliance. Importantly, please provide a point-by-point response and accordingly text changes, further discussion, acknowledgment of weaknesses as well as clarifications. Some more info, statistical analyses and altered figure representation would be needed to address the reviewers' specific points. The causal link between computational and experimental data could be addressed by analysing the data at hand to show that the protein you study experimentally is representative of a trend you are reporting in your computational analysis.

Thank you for this interesting contribution to Life Science Alliance. We are looking forward to receiving your revised manuscript.

REFEREE REPORTS OBTAINED DURING PEER REVIEW ELSEWHERE

Reviewer 1

This study first analyses data (from literature and databases) to understand the degree to which methylarginine is clustered (or not clustered) in the human proteome. It then analyses one protein with clustered RG motifs in detail, being SYNCRIP, to understand the function of the clustered methylarginines therein. Particular focus is on the question of whether the clustered methylarginines provide a different, tunable regulatory mechanism for the methylation writer (PRMT1) and the reader SMN1.

A question I have, which the Editors will be able to help with, is of suitability of this paper for this journal. As noted above, data is shown to build a model that the disordered regions in proteins like SYNCRIP, which carry the disordered RG domains, can act as tunable regulators. And it is suggested but not shown that this mechanism is likely to be used in many proteins in the proteome. What I am not sure about is whether these main results make the study a good fit for this journal or whether it might fit better with a protein or biochemistry journal? Whilst screens are used to find interactors of SYNCRIP, the interactor SMN1 itself is not novel here. However the detailed studies of the interactions are of novelty (but focused on one protein rather than many in the proteome or interactome).

Major considerations:

The data showing the existence of clustered methylarginine is of some but not absolute novelty. PRMT1 is known to act on stretches containing repeats of RGs and its yeast ortholog is well known to act on tandem repeats of RGGs. What was not necessarily apparent in the study was whether just PRMT1 substrates have RG clusters or whether these could also be sites for other PRMTs? A motif analysis of the sites in and not in clusters would help firm up an answer to this question. RG or RGG is a clear motif for PRMT1 and other motifs for other PRMTs have been suggested in proteome-scale analyses of human methylation.

The authors have not discussed the role of dimethylation in their study or model. This is an

important consideration as each R in an RG or RGG can be subject to mono or (asymmetric)dimethylation. PRMT1, given it is a partially but not completely processive enzyme, can methylate the same site once or twice. This means that PRMT1 can and will interact with non-methylated R but also monomethylated R. Tudor domains interact predominantly with dimethylarginines. These issues need to be considered and addressed in the models presented throughout. The issue of mono or dimethylation is further complicated in the study in that the western blotting, for the detection of methylation, has been done with an un-named anti-pan-methylarginine antibody. One assumes that this detects mono and asymmetric di-methylation (and hopefully not symmetric di-methylation). So the methylation status of SYNCRIP in Figure 3A and B could be either mono or di-methylation. In other experimentation, it will also be unknown whether mono or di (or more likely a mix of) methylation has occurred. Mass spectrometry or use of mono or di-methylation-specific antibodies could be useful here.

The paper makes an argument that methylation in extended disordered regions with RG motifs is part of a regulatory process. I agree that this is likely. However whilst it has been shown that more or less SMN1 can bind to SYNCRIP with more or less methylatable R, I'm not completely sure that it has been proven that this happens as part of an *in vivo* regulatory event. This affects the argument and overall conclusion that a functional, tunable process exists.

Investigations into the functional relevance of the methylated SYNCRIP show promise but are not very conclusive (Figure 3D, 3E). The absence of SMN1 as an interactor makes it unclear how important the interaction of SMN1 to SYNCRIP is for the biology of stress granules. Unfortunately this also casts a shadow onto the biological importance of the arginine methylation for the SMN1 interaction (given that the SYNCRIP is expected to be methylated and thus SMN1 should be an interactor). So it is probably inappropriate, at the start of the 2nd paragraph to say "Having validated the functional relevance of the SYNCRIP C terminal methylation in cells...". The sentence needs to be changed to be more balanced given that the functional relevance has not been proven. Text elsewhere that assumes the SMN1-SYNCRIP interaction is methylation dependent also needs to be carefully reviewed.

Minor considerations:

Abstract.

The first sentence is obtuse and hard to understand; it should be reworded.

The last sentence suggests that the cellular function of the di-amino acid motifs has been addressed in the paper. Given the lack of interaction between SMN1 and SYNCRIP *in vivo* (and thus function has not been proven), I am not sure that one can say that SYNCRIP is an example of how "PTMs leverage ... disordered sequences to drive cellular functions".

Page 8.

I think there are a few references to the wrong figure here? Should that be references to middle and right panels of 1H?

Page 12. 2nd paragraph.

"Together these experiments provide further evidence that SYNCRIP is methylated under basal conditions in HEK293T cells and that this methylation leads to direct binding of SYNCRIP to SMN1."

Has the latter part of this sentence (methylation leads to direct binding) actually been shown? Fig 3C does not prove that, as there is no evidence that methylation is required for that interaction. Addition of a methylation inhibitor to the split-EYFP experiment would better

prove this statement.

Transfer Review:

Yes, my report and identity can be transferred confidentially

Reviewer 2

General comments

The study examines the potential roles of arginine methylation in disordered regions across the proteome using publicly available data. The authors particularly focus on HnRNP SYNCRIP. Using protein interaction assays, the role of arginines and their arrangements is functionally assessed. The authors establish functional significance of disordered regions composed of RG motifs, in the context of SYNCRIP, methyl-binding protein SMN1 and methyl transferase PRMT1.

The study lacks originality at the broadest scale and does not report comprehensive analysis. The interest for systems biologists is expected to be limited. In addition, the writing could be significantly improved in some sections. Most of the novel observations derive from the small-scale experiments and relatively little new is learned at the systems level.

Major comments

1. Novelty in findings.

The methylation data is a reanalysis of previously published data and the analysis are generally a confirmation of what has been observed for other PTMs in disordered regions. Any PTMs in these regions are therefore likely to evolve faster (see PMID: 29858274 and many other papers). The analyses here are therefore rather confirmatory. There seems to be many things known on the binding strength of the tudor domain. For example, it is dependent on the hydrophobic methyl groups. Therefore the interaction with SYNCRIP is expected to increase with the number of methyl arginines present in the disordered region. Also, it has been discussed before that strength of interaction with SYNCRIP increases with the total number of methyl arginines present in the disordered region (PMID: 20937909,22363433 and 22101937).

2. Extent of advantage over peptide based in vitro assays.

In terms of the methodology, some novelty of the study lies in the fact that full-length proteins rather than peptide-based assays are used. However, for the immunoprecipitation and lumier assays, the proteins were extracted out of the cell, which would qualify the assay an in vitro assay rather than an in vivo assay. Also, as the regions studied are structurally disordered, one would argue that their properties are likely to be very similar whether they are contained in a protein scaffold or studied in isolation. Therefore, relative to peptide based in vitro assays, the extent of advantage offered by the core methodology used in the study may be limited. The claims suggesting that the study captures in vivo scenario may need to be adjusted (at page 4 line 14 and at page 16 line 16).

3. Expression and mis-sense variation analysis

The use of expression analysis in the context of clustering prevalence is not well defined both in the main text and in methods. The same data is referred to in different ways, which may be misleading. The text refers to it as "ratio of samples" (page 8 line 1), the figure says "ratio of genes" and the figure caption says "overexpression ratios". Additionally, the expression analysis would only 'suggest' rather than 'highlight' the role of gene regulatory signatures (Page 8 line 3).

In case of both expression and mis-sense variation analysis, the readability is also confounded by mislabeling of the subplots and panels in Figure 1. See my related minor comments below. For instance, there is no middle or right panel in Figure 1G.

4. Underdeveloped but important aspects.

(1) Arginine mono-methylation vs multi-methylation.

This aspect is completely untouched throughout the manuscript. In the 'Experimental procedures', I could not find any description about whether the 'pan-methylated-arginine antibody' which is used to detect methylation of SYNCRIP binds to monomethylated or dimethylated arginine.

Also, this aspect should be acknowledged while referring to Larsen et al. 2016 study at Page 3 Line 15. The sentence would instead read ".. at least 7% of arginine sites in the expressed proteome are "mono"-methylated (Larsen et al. 2016). ..".

(1) RNA binding.

Even though SYNCRIP is a RNA binding protein, and RGG motif is known to interact with RNA (PMID 21642970), the RNA binding aspect is underdeveloped in the manuscript. To evaluate the influence of RNA binding on the Arginine methylation signaling, the authors could carry out the immunoprecipitation assays in presence of RNA substrates of SYNCRIP.

(3) Implications of the roles of hnRNP and SMN1 in stress granule formation are understated and are not followed up in the discussion section. The authors do not observe the recruitment of SMN1 to stress granules similar to (Guil et al. 2006; Zou et al. 2011) studies. An explanation for this unexpected result is missing.

A study on similar protein : TDP-43 hnRNP containing Low-Complexity C-Terminal Domain with RGG motifs found that RNA binding plays a major role in phase separations, owing to RGG motif-RNA binding. (PMID: 27545621). The manuscript could possibly benefit from extending on such strands of evidence. The authors could monitor whether the different mutant forms of RGG motifs are recruited to the stress granules.

(4) Glycines

In the abstract page 2 line 11, the authors claim that the findings establish the roles of "di-amino" motifs. However, in the manuscript, they only discuss about arginines in the tail of SYNCRIP.

For instance, the authors report that the modified arginines function in concert irrespective of their position within disordered region (page 5 lines 11-12). Here the authors could carry out window based sequence analysis for glycines to see if there are any patterns in positions/arrangements of glycines that could explain the result.

5. Distinct or overlapping?

According to the manuscript, distinct modes of signaling mechanisms of arginine methylation are based on

- (1) total number of arginines in the disordered region and
- (2) density of methyl arginines within the disordered region.

To me, these two aspects seem overlapping rather than distinct. In the abstract (Page 2 line 1), instead of '.. arginine methylation events bifurcates signaling mechanism ..', the authors should have posed that '.. arginine methylation events provide dual modes of signalling mechanism ..'.

Minor comments

1. Cumulative regulatory potential

From the abstract, it is not clear what the authors mean by 'Cumulative regulatory potential'. The readers have to come to page 9 lines 8 to 10 to understand the meaning. As one of the main findings of the study, I think it should be explained in the abstract itself.

2. Page 3 Line 5

The review Chen et al 2011 is majorly discussing arginine methylation, but not general PTM system. PMID 25692714 could be an appropriate review for providing background of the field to the general audience.

3. Page 5 Line 5.

"panel of 37 full length proteins" is misleading.

If anything it is a rearrangement of RGG domains along the C terminal disordered region of SYNCRIP.

4. Page 6 line 7

In Figure 1B, sequence related features were analyzed rather than biophysical ones.

Also, the title of figure caption describes the analysis as "biochemical" which is misleading.

5. page 6 line 8

In order to provide a full context, the authors could mention the features of the other 70% of the arginine methylation sites. Also, this point could be carried to the Discussion section to explain whether the findings of the study are in any way informative in understanding the mechanisms of the remaining 70% of the Arginine methylation sites.

6. Page 6 line 11

It is quite obvious that if there were more methyl Rs per 20 amino acid window, it is more likely that there would be more methyl Rs per RG motifs. The analysis from figure 1B could be presented as a supplementary information.

7. Page 13 line 17

It would be combinations rather than permutations, as the order does not matter. They are all Arginines. Therefore the number of possibilities would be lower than what is estimated.

If L is the length of the C terminal tail of SYNCRIP, combinations are $L!/(19! \times (L-19)!)$

8. The meaning of the sentences is quite unclear and would need improvement.

- (1) page 8 lines 20 to 22

Arginine is the identity of the amino acid.

(2) page 11 lines 5 to 7
Confusing and repetitive.
(3) page 16 line 5

9. Typos

Page 5 line 17

It's PhosphoSitePlus rather than phosphositeplus.

page 10 line 24

By HNRP, the authors are probably referring to "heterogeneous nuclear ribonucleoproteins (hnRNP)".

10. Commas

Inserting commas between long sentences would improve the readability of few sentences.
e.g. page 2 line 13

11. Page 8 line 20

Full form of the abbreviation Me-R should be declared.

12. No section titles in the results.

13. Error in labeling the subplot.

(1) Figure 1C

The key to the symbol 'D' is missing in.

(2) Figure 1F

Mentioned on page 8 line 3:

location of the panel should be indicated. Rightmost in this case.

(3) Figure 1G

Mentioned on page 8 line 20: there is no middle or right panel.

(4) Figure 2C

It is mislabeled (as B) in the figure caption.

(5) Figure 3B

There is no middle panel.

14. As the C terminal tail of SYNCRIP is approximately 100 amino acid long, the readers would benefit from visualizing the protein sequence in figures 1E, 2A and 4A.

In addition to the description in the text, for example: page 11 lines 8 to 9, a schematic with actual protein sequence would be more informative.

15. Statistical testing

Wherever Mann-Whitney-Wilcoxon tests are used, in figure captions, it should be mentioned whether the test is one-sided or two-sided.

16. Page 6 line 14

Wherever applicable, references to the methods sections should be included.

17. Page 20 line 4

Authors could have mentioned the overlap length. It could be an overlap of 5 residues leading to sampling bias. Ideally there would be an overlap of 1 residue.

18. Structural analysis

Because it is a structural analysis, it would be useful to add a crystal structure, if available, in the supplementary information indicating location of methyl binding and which residues are mutated.

19. Western blot

Figure 3A

Quantitative data should be represented with statistical tests applied over biological replicates. The gel pictures (whole rather than cut-out/cropped) can be presented in supplementary information.

20. Image analysis

For Figure 3C

Authors could have shown an overlap of Hoechst and GFP filters. That would make the result clearer.

Figure 3C and 3E

It would be better to substantiate the results related to colocalization experiments with appropriate statistical tests (such as in PMID: 24117417) applied.

21. In case of 'GTEx dataset' and 'Gnomad data analysis', the websites and the date of accession should be mentioned in Methodology section.

22. Molecular docking.

In support of the major claim of the study that seemingly random positioning rather than contiguous arrangements of arginines allows better binding, the authors could try to simulate the binding using molecular docking. Otherwise, it is unclear why this would happen. As binding of proteins is the core subject of the study, a detailed structural analysis could provide valuable depth of information.

23. Page 2 line 6

The manuscript may benefit from extending on the lines of "tunable interaction surface". The aspect is largely untouched in the introduction and the discussion. There is evidence (PMID: 21868366) that alternate splicing generated isoforms that allow for the presence or absence of the motifs, providing the tunability. The manuscript would benefit from such a discussion of such events if they occur in the case of SYNCRIP.

24. Mutation rate and rate of evolution (or degree of polymorphism) at specific positions across the genome are different things. Mutations are the direct result of the process of error incorporation in the genome, not how much diversity there is in a population. Saying that some sites are more prone to mutations is therefore incorrect because it suggests that they have higher mutation rates. Most likely they do not but rather mutations at these sites are less likely to be removed by natural selection.

Reviewer 3

Woodsmith et al., computationally analyse the publicly available data on arginine methylation (R-methyl) sites in human proteins and show that (i) R-methyl sites tend to cluster within disordered regions of proteins, and that the clustering is more pronounced in the presence of 'RG' motifs and (ii) such R-methyl clusters are enriched in RNA-binding proteins (RBPs). By integrating R-methyl data with human tissue-wide gene-expression data (GTEx), they show that genes that contain R-methyl clusters are low-abundant and have less expression

variability, and interpret that the expression of these genes is tightly regulated. By analyzing human natural variation data (gnomAD), the authors show that R-methyl clusters, compared to isolated R-methyl sites, tend to accommodate more mutations and reason that an array of R-methyl sites, could compensate for the loss of individual modification sites. By experimentally investigating R-methyl cluster in the multiple 'RG' motif containing RBP SYNCRIP, they identify that (i) SMN1 and PHF1 bind to SYNCRIP in R-methylation and K-methylation dependent manner, (ii) PRMT1 methylates SYNCRIP R-sites, (iii) R-methylation is required for efficient recruitment of SYNCRIP to stress granules, though SMN1 recruitment could not be detected and (iv) modified and unmodified 'RG' motifs mediated interactions cumulatively. Furthermore, using mutagenesis studies, they show that SMN1 binding is affected by total arginine content while PRMT1 binding required contiguous arginine stretches and propose that there could be two different modes of binding within these regions. The manuscript is written well and the figures are elegantly presented.

While some of the results presented here are correlative, the study in general does not offer broad insights pertaining to R-methylation or deep insights regarding SYNCRIP interaction(s) and functions. We recommend reconsideration of this work upon major revisions as discussed below:

Major Comments

1. There is no clear demarcation between the main findings and the peripheral ones. This is a result of the general lack of interpretations and inadequate connection between different findings and a clear overall take-home message. It is not clear as to how the experimental findings build on computational investigations to advance our understanding. Furthermore, the validity of generalizing the interpretations on one protein to a proteome-scale is questionable (specifically claims in the fourth paragraph of Discussion section).
2. The connection between computational and the experimental parts is weak. A figure that shows the overlap between the proteins with high arginine methylation density with R-methyl transferase interaction could serve this purpose and render the selection of SYNCRIP more meaningful. Nevertheless, the authors need to connect experimental findings with computational results and provide concrete biological insights.
3. Computational investigations: (i) Effect size estimates for the observed differences need to be provided for each comparison. (ii) Are the trends in Fig 1A-1C statistically significant? (iii) Enrichment of R-methyl clusters in disordered proteins, RBPs, tightly controlled proteins, and those with more mutations are all isolated observations. How are these observations interlinked and what is the connection with R-methyl clusters. In other words, there is no correlative or causal evidence to suggest that the observed trends are cause or consequence of clustered R-methyl sites. (iv) Tight regulation (Fig. 1F): The analyses show that proteins with R-methyl clusters are low-expression genes and have low-variability across different tissues. How this inform about tight regulation, as there is no analyses on synthesis or degradation rates? Even if it does, disordered proteins are known to be tightly regulation, R-methyl clusters are enriched in disordered proteins and hence one would expect this correlation. Why is this surprising or an important finding? As pointed earlier, the authors should dwell on the possible connection(s) of R-methyl clusters and the observed trends. In addition, they should consider comparing proteins with dispersed (non-clustered) R-methyl sites with those that contain clustered R-methyl sites. Furthermore, the observations could be confounded by the differences in the total no. of tissues analyzed for proteins with R-methyl clusters and the controls. (v) It appears that the authors try to motivate their choice of SYNCRIP by looking for enrichment of only RBPs in proteins with R-methyl clusters. While this might be valid, a simple Gene Ontology enrichment analysis will provide information about other functional classes that are enriched in both in proteins with clustered and dispersed R-methyl sites. (vi)

How was the 20 amino acid window determined? How will the results change at different window sizes?

4. Experimental investigations: The outcome of the experimental investigation is convoluted. (i) Only 15 of the 19 R-sites investigated, are known to be methylated. PRMT1 binding does not require methylated arginines (Fig. 3B). Though SMN1 requires R-methyl sites for binding, it is not clear which non-mutated R-sites are methylated. Therefore, it is not clear if the observed effects are because of clustered arginines or clustered 'RG' motifs or clustered 'R-methyl sites'. (ii) What is the rationale behind selecting the sites that were mutated in 6Rs and 14Rs? (iii) Clustering the heatmaps in Fig. 4A based on the binding signal (SMN1 and PRMT1) might be more helpful to interpret the results and follow the description. Details of the controls used in Fig. 4B-4C needs to be provided. (iv) No statistical significance assessment or any statistical metric is provided for Fig. 4D-4G. Therefore, it becomes a bit difficult to assess the importance of these findings. Furthermore, in Fig.4D-4E, the bin sizes are variable between the different R-mutants. Please consider providing continuous scatter plots to avoid such variability in bin sizes. (v) The R-mutant bins are again misleading here. For instance, the 10-13 bin essentially includes all the datapoints ranging from 10-12 considered in the 9-12 bin. Would generating distinct bins or plotting this data on continuous scales, lead to the same interpretations?

Minor Comments

5. The focal point of this study appears to be 'binding potential of R-clusters' (well presented in Fig. 5). Therefore, the title and the repeated discussion about 'cumulative regulatory potential' is misleading and often confusing. What regulation are the authors referring to?

6. Usage of semantics: Low structural complexity and low-complexity sequences have been used synonymously. The authors have not computed any sequence-complexity measures (such as those computed by SEG) and should consider using disordered regions instead of low-structural complexity or low complexity. Similarly, phrases such as regulatory capacity, regulatory potential and regulatory interactions, need to be explicitly defined or avoided. As such none of the experiments done investigate the regulatory nature of the R-methyl sites. Please consider using non-stress instead of basal for sub-cellular localization experiments.

7. The Results section should be divided into sub-sections with the overall finding/conclusion of each sub-section provided as title, for better comprehension of the findings.

8. Residue numbers, such as R409 and R411, used to represent 'R' mutants (in 6Rs and 14Rs) are difficult to follow as there is no reference provided in the manuscript. Please consider providing the residue numbers in Supplementary Figure 1C. Alternatively please use 3-letter code (e.g. Arg-409, etc.)

9. The last paragraph in the introduction section about di-amino acid repeats is abrupt, as there is no clear logical flow from R-methylation, R-content to di-amino acid repeats.

10. Ratio in the y-axis label of Fig. 1A, 1B and 1D should be proportion or fraction. No. of datapoints in each distribution needs to be mentioned. In Fig 4D-4G, should the y-axis label be Log2 binding signal?

11. The distribution profile of disorder for each amino acid across the length of SYNCRIP should be provided as a supplementary figure.

1st Authors' Response to Reviewers: September 5, 2018

Woodsmith et al. point by point reply 1

Point by point reply, with answers highlighted in blue

Reviewer #1:

This study first analyses data (from literature and databases) to understand the degree to which methylarginine is clustered (or not clustered) in the human proteome. It then analyses one protein with clustered RG motifs in detail, being SYNCRIP, to understand the function of the clustered methylarginines therein. Particular focus is on the question of whether the clustered methylarginines provide a different, tunable regulatory mechanism for the methylation writer (PRMT1) and the reader SMN1.

A question I have, which the Editors will be able to help with, is of suitability of this paper for this journal. As noted above, data is shown to build a model that the disordered regions in proteins like SYNCRIP, which carry the disordered RG domains, can act as tunable regulators. And it is suggested but not shown that this mechanism is likely to be used in many proteins in the proteome. What I am not sure about is whether these main results make the study a good fit for this journal or whether it might fit better with a protein or biochemistry journal? Whilst screens are used to find interactors of SYNCRIP, the interactor SMN1 itself is not novel here. However the detailed studies of the interactions are of novelty (but focused on one protein rather than many in the proteome or interactome).

Major considerations:

The data showing the existence of clustered methylarginine is of some but not absolute novelty. PRMT1 is known to act on stretches containing repeats of RGs and its yeast ortholog is well known to act on tandem repeats of RGGs. What was not necessarily apparent in the study was whether just PRMT1 substrates have RG clusters or whether these could also be sites for other PRMTs? A motif analysis of the sites in and not in clusters would help firm up an answer to this question. RG or RGG is a clear motif for PRMT1 and other motifs for other PRMTs have been suggested in proteome-scale analyses of human methylation.

In our initial analysis we identified clusters that were driven by RG motifs and clusters that were driven by non-RG motifs (Figures 1H and Supplementary Figure 1B). In order to determine whether any specific PRMT can be assigned to clusters defined by non-RG motifs, we performed a motif analysis on all non-RG methylation sites in these clusters (details in materials and methods). We identified a motif with qualitative similarity to those identified as CARM1 substrates by Shishkova et al. Nat Comms. With the central arginine flanked at various positions by a proline (now Supplementary Figure 1B). As such we have added this analysis and clarified this distinction in the text.

The authors have not discussed the role of dimethylation in their study or model. This is an important consideration as each R in an RG or RGG can be subject to mono or (asymmetric)dimethylation. PRMT1, given it is a partially but not completely processive enzyme, can methylate the same site once or twice. This means that PRMT1 can and will interact with non-methylated R but also monomethylated R. Tudor domains interact predominantly with dimethylarginines. These issues need to be considered and addressed in the models presented throughout. The issue of mono or dimethylation is further complicated in the study in that the western blotting, for the detection of methylation, has been done with an un-named anti-pan-methylarginine antibody. One assumes that this detects mono and asymmetric di-methylation (and hopefully not symmetric di-methylation). So the methylation status of SYNCRIP in Figure 3A and B could be either mono or di-methylation. In other experimentation, it will also be unknown whether mono or di (or more likely a mix of)

methylation has occurred. Mass spectrometry or use of mono or di-methylation-specific antibodies could be useful here.

We agree with the reviewer that the actual methylation state of methylated R is a subject of importance. However we want to outline that the data on this subject are largely unresolved in the literature and therefore our bioinformatic analysis of these data cannot contribute to clarification of this topic. If we look at the most prominent papers in the literature (referred to in the introduction first paragraph) that contributed mass spec based Me-R identifications it is quickly clear that differences of the methylation states were not considered:

Larsen et al. *Sci Signaling*: “Because monomethylation is a prerequisite for the enzymatic catalysis of dimethylation, identification of monomethylated arginine residues is a strong proxy for arginine methylation sites.” and “The stoichiometry is derived from an intricate interplay between the activities of enzymes capable of adding and removing the PTM, along with the overall turnover of the modified protein substrate. For arginine methylation, this is further complicated by the fact that monomethylation may be further converted into dimethylation, which can affect its stoichiometry;”

Geoghegan et al. *Nat Communications*: “Reflecting the Abs specificity, the majority of methylation sites were identified as MMA (Supplementary Fig. 1d), which is considered to be an intermediate methylation state towards di-methylation” and “Previous studies have provided convincing evidence that mono-methylated arginine is often a transition state toward di-methylation and it is likely that most methylations sites we found are transition states”

Bremang et al. *Mol Biosyst*: This study describes the use of six antibodies with targeted specificity against mono(MMA), di, sym-di and asym-di- (ADMA) Me peptides and identified a total of 254 methylated R residues. One antibody did result one site only in the IP experiments. The other five antibodies together yielded 88 sites specific to one antibody only, while 112 were in the overlap. They further observed: “On arginine residues, the majority of sites were di-methylated (210; 65%) – an observation that is in line with previous reports (Fig. 5D). A large proportion of these sites was also observed in the mono-methylated state, raising similar conclusions on the processivity of the arginine methyltransferases (PRMTs).”

Guo et al. *Mol Cell Proteomics*: In this seminal study two MMA and two ADMA antibodies were used for methyl-R enrichment. Many more MMA sites than ADMA sites were detected. However MMA and ADMA are highly coupled, the authors show that under their conditions, ADOX treatment decreased ADMA and strongly increased MMA immunoreactivity. Notably the extend of overlap between the two ASMA antibodies (JJ: 0.13) is almost identical to the overlap between MMA and ADMA (JJ: 0.11) identifications.

In summary, the current literature does report many more MMA sites than ADMA sites, MMA is viewed as a general measure for arginine methylation because the two types of modifications seem highly coupled. Overlap between MMA sites and ADMA, even at low sampling depth, is high. The major bias towards MMA stems from the fact that MMA can be detected on classical tryptic peptides, while ADMA prevents tyrpsin from cleavage and thus larger and differently charged peptides with distinct properties need recording for reliable ADMA identification (Hart-Smith G, *Mol Cell Proteomics*. 2016). On the other hand, quantitative comparison of MMA and ADMA via mass spec for the same arginine sites in particular in repeat windows like the one addressed here is currently prohibitively difficult (comparisons of a tryptic peptide with a miss-cleaved version). Therefore our computational and experimental analyses do not resolve MMA and ADMA and we refer throughout the manuscript to arginine methylation.

For the specific suggestion of the reviewer to probe SYNCRIP with different antibodies we are facing the same problems outlined, dealing with a window of a series of methylated arginine. 8 of the arginines in the C terminal tail of SYNCRIP have been observed in both the mono- and di-methylated state, 6 only in mono- and 1 only in di-state. Therefore, as the reviewer suggests in brackets we are, to all likelihood, facing a mix of different methylations at many sites that cannot be distinguished with antibodies. Since antibodies would only report a qualitative signal over the whole protein (as the one we report with the pan-ME antibody) we cannot quantify MMA against ADMA and resolve this information. As such the use of different antibodies does not contribute to any finding or conclusion drawn in the paper. We used an anti-mono and dimethyl arginine antibody (AbCam, [7E6], ab412, raised against asymmetrical NG/NG-dimethyl arginine) now fully described in the material methods section.

The paper makes an argument that methylation in extended disordered regions with RG motifs is part of a regulatory process. I agree that this is likely. However whilst it has been shown that more or less SMN1 can bind to SYNCRIP with more or less methylatable R, I'm not completely sure that it has been proven that this happens as part of an in vivo regulatory event. This affects the argument and overall conclusion that a functional, tunable process exists.

We have altered the title, abstract and discussion to focus more on the methylation dependent interaction results rather than a broader functional implication.

Investigations into the functional relevance of the methylated SYNCRIP show promise but are not very conclusive (Figure 3D, 3E). The absence of SMN1 as an interactor makes it unclear how important the interaction of SMN1 to SYNCRIP is for the biology of stress granules. Unfortunately this also casts a shadow onto the biological importance of the arginine methylation for the SMN1 interaction (given that the SYNCRIP is expected to be methylated and thus SMN1 should be an interactor). So it is probably inappropriate, at the start of the 2nd paragraph to say "Having validated the functional relevance of the SYNCRIP C terminal methylation in cells...". The sentence needs to be changed to be more balanced given that the functional relevance has not been proven. Text elsewhere that assumes the SMN1-SYNCRIP interaction is methylation dependent also needs to be carefully reviewed.

We have presented data that to show that the SMN1 SYNCRIP interaction is methylation dependent in cell culture, both using a methylation inhibitor (Figure 3A) and our mutant panel that show reduced methylation levels (Figure 4 and Supplementary Figure 6), as well as Tudor domain mutants in SMN1. As such we do not think we this statement in the text is well supported. While a function of SYNCRIP methylation in the cell is to recruit SMN1, we acknowledge that the broader function of SYNCRIP and SMN1 together in the cell driven by this methylation is a weaker point in the manuscript (mainly due to problems in detection of endogenous SMN1). Please note that this is also stated clearly as a negative result in the text and presented in Supplemental Figure 4. However, we have altered the title, abstract and discussion to focus more on the methylation dependent interaction rather than a broader functional output.

Minor considerations:

Abstract.

The first sentence is obtuse and hard to understand; it should be reworded.

Altered in text to make more clear.

The last sentence suggests that the cellular function of the di-amino acid motifs has been addressed in the paper. Given the lack of interaction between SMN1 and SYNCRIP in vivo (and thus function has not been proven), I am not sure that one can say that SYNCRIP is an example of how "PTMs leverage ... disordered sequences to drive cellular functions".

Altered to “cellular interactions” instead of “cellular functions”.

Page 8.

I think there are a few references to the wrong figure here? Should that be references to middle and right panels of 1H?

Altered in new text.

Page 12. 2nd paragraph.

"Together these experiments provide further evidence that SYNCRIP is methylated under basal conditions in HEK293T cells and that this methylation leads to direct binding of SYNCRIP to SMN1." Has the latter part of this sentence (methylation leads to direct binding) actually been shown? Fig 3C does not prove that, as there is no evidence that methylation is required for that interaction. Addition of a methylation inhibitor to the split-EYFP experiment would better prove this statement.

We provided support for the methylation dependency in three different ways: 1) Tudor domain mutants known to prevent methylation dependent interactions (Figure 2). 2) Methylation on SYNCRIP decreases in the mutants which also have lower SMN1 binding (Figure 3A, Figure 4 and Supplementary Figure 6) and finally 3) in Figure 3A, as the reviewer suggested we had used a methylation inhibitor to disrupt binding, here in a pull down experiment. Of note, split EYFP binding is not reversible so would not likely be a suitable system to test binding in the presence of a methylation inhibitor.

Reviewer #2:

General comments

The study examines the potential roles of arginine methylation in disordered regions across the proteome using publicly available data. The authors particularly focus on HnRNP SYNCRIP. Using protein interaction assays, the role of arginines and their arrangements is functionally assessed. The authors establish functional significance of disordered regions composed of RG motifs, in the context of SYNCRIP, methyl-binding protein SMN1 and methyl transferase PRMT1.

The study lacks originality at the broadest scale and does not report comprehensive analysis. The interest for systems biologists is expected to be limited. In addition, the writing could be significantly improved in some sections. Most of the novel observations derive from the small-scale experiments and relatively little new is learned at the systems level.

Major comments

1. Novelty in findings.

The methylation data is a reanalysis of previously published data and the analysis are generally a confirmation of what has been observed for other PTMs in disordered regions. Any PTMs in these regions are therefore likely to evolve faster (see PMID: 29858274 and many other papers). The analyses here are therefore rather confirmatory. There seems to be many things known on the binding strength of the tudor domain. For example, it is dependent on the hydrophobic methyl groups. Therefore the interaction with SYNCRIP is expected to increase with the number of methyl arginines present in the disordered region. Also, it has been discussed before that strength of interaction with SYNCRIP increases with the total number of methyl arginines present in the disordered region (PMID: 20937909,22363433 and 22101937).

The pubmed IDs pointed out above are relevant, two of which we referenced in our original text. They representing short peptide biophysical studies using distinct TUDOR domains, none does deal with SYNCRIP. As peptides studies can not address the large regions we systematically identified in our bioinformatic analysis, several hypotheses could be made as to the function of many arginine methylation events in these regions (redundancy, used in distinct contexts, used for multiple PRTMs etc). Not only are we the first to systematically identify these regions for arginine methylation, we provide the first experimental evidence suggesting a function for such extreme PTM events (see below for advantages of experimental system).

2. Extent of advantage over peptide based in vitro assays.

In terms of the methodology, some novelty of the study lies in the fact that full-length proteins rather than peptide-based assays are used. However, for the immunoprecipitation and lumier assays, the proteins were extracted out of the cell, which would qualify the assay an in vitro assay rather than an in vivo assay. Also, as the regions studied are structurally disordered, one would argue that their properties are likely to be very similar whether they are contained in a protein scaffold or studied in isolation. Therefore, relative to peptide based in vitro assays, the extent of advantage offered by the core methodology used in the study may be limited. The claims suggesting that the study captures in vivo scenario may need to be adjusted (at page 4 line 14 and at page 16 line 16).

Synthesising dozens of PTM modified peptides over 100 amino acids in length, as used here in our experiment set up, is technically improbable and extremely expensive. The main advantage provided by our setup is, for the first time, being able to address the function of such expansive modified protein sequences. Furthermore, we can show that in a cellular context, these sequence can be modified and bound by regulatory proteins. The genetic approach to Me-R-recognition has not been taken for any protein before, and even short peptide level experiment for SYNCRIP are lacking from the literature. Figure 4 is the most comprehensive binding analysis of any kind of such PTM cluster region. To make this novelty more clear in the text, we now refer to the extensively modified arginine regions found as arrays, we which also experimentally test in the form of full length SYNCIRP. We also modified the statements criticized by the reviewer.

3. Expression and mis-sense variation analysis

The use of expression analysis in the context of clustering prevalence is not well defined both in the main text and in methods. The same data is referred to in different ways, which may be misleading. The text refers to it as "ratio of samples" (page 8 line 1), the figure says "ratio of genes" and the figure caption says "overexpression ratios". Additionally, the expression analysis would only 'suggest' rather than 'highlight' the role of gene regulatory signatures (Page 8 line 3).

Altered the text / figures as suggested

In case of both expression and mis-sense variation analysis, the readability is also confounded by mislabeling of the subplots and panels in Figure 1. See my related minor comments below. For instance, there is no middle or right panel in Figure 1G.

Altered the text / figures as suggested

4. Underdeveloped but important aspects.

(1) Arginine mono-methylation vs multi-methylation.

This aspect is completely untouched throughout the manuscript. In the 'Experimental procedures', I could not find any description about whether the 'pan-methylated-arginine antibody' which is used to detect methylation of SYNCRIP binds to monomethylated or dimethylated arginine.

The antibody is now described in the methods section. Please see also the answer to reviewer 1.

Also, this aspect should be acknowledged while referring to Larsen et al. 2016 study at Page 3 Line 15. The sentence would instead read ".. at least 7% of arginine sites in the expressed proteome are "mono"-methylated (Larsen et al. 2016). ..".

Altered in text.

(2) RNA binding.

Even though SYNCRIP is a RNA binding protein, and RGG motif is known to interact with RNA (PMID 21642970), the RNA binding aspect is underdeveloped in the manuscript. To evaluate the influence of RNA binding on the Arginine methylation signaling, the authors could carry out the immunoprecipitation assays in presence of RNA substrates of SYNCRIP.

We do not address the RNA binding functions of SYNCRIP in the manuscript. Though this is an important aspect as well as protein binding and could be an additional line of research, this aspect does not contribute to any finding or conclusion presented in the manuscript.

(3) Implications of the roles of hnRNP and SMN1 in stress granule formation are understated and are not followed up in the discussion section. The authors do not observe the recruitment of SMN1 to stress granules similar to (Guil et al. 2006; Zou et al. 2011) studies. An explanation for this unexpected result is missing.

Guil et al refers only to the recruitment of hnRNPs to stress granules. It appears that recruitment to stress granules is concentration dependent. Reduction in SMN1 expression in PC12 cells reduced SG formation and its recruitment (Zou et al). We observed SMN1 recruitment to SG upon over expression, but not when we used antibodies to stain for endogenous protein. Therefore we didn't include the overexpression data in the final report, but report the negative results with endogenous SMN1.

A study on similar protein : TDP-43 hnRNP containing Low-Complexity C-Terminal Domain with RGG motifs found that RNA binding plays a major role in phase separations, owing to RGG motif-RNA binding. (PMID: 27545621). The manuscript could possibly benefit from extending on such strands of evidence. The authors could monitor whether the different mutant forms of RGG motifs are recruited to the stress granules.

In the original manuscript we did observe lower recruitment of a SYNCRIP mutant containing fewer arginines to stress granules (Figure 3E). The focus of the manuscript is however on the impact of the (methylated) arginines on protein interactions with SMN1/PRMT1, and not on stress granules per se. We dissected the protein interactions in detail using distinct mutants and just used two versions, wild type and 6R mutant, to exemplify the impact on stress granule recruitment. One of the co-authors indeed plans to investigate the suggested direction.

(4) Glycines

In the abstract page 2 line 11, the authors claim that the findings establish the roles of "di-amino" motifs. However, in the manuscript, they only discuss about arginines in the tail of SYNCRIP. For instance, the authors report that the modified arginines function in concert irrespective of their position within disordered region (page 5 lines 11-12). Here the authors could carry out window based sequence analysis for glycines to see if there are any patterns in positions/arrangements of glycines that could explain the result.

18 of the 19 arginines in the C terminal tail of SYNCRIP are in "RG" motifs (Figure 1B,Supplementary Figure 1), meaning that they are immediately followed in the protein sequence by a glycine, hence the reference to a di-amino acid motif. That said, we removed this phrase from the manuscript to provide more clarification to what

we are referring to. As almost all arginines are accompanied by a glycine, it is not possible to scan them for patterns that would be independent from any arginine patterns we observe.

5. Distinct or overlapping?

According to the manuscript, distinct modes of signaling mechanisms of arginine methylation are based on (1) total number of arginines in the disordered region and (2) density of methyl arginines within the disordered region.

To me, these two aspects seem overlapping rather than distinct. In the abstract (Page 2 line 1), instead of '.. arginine methylation events bifurcates signaling mechanism ..', the authors should have posed that ".. arginine methylation events provide dual modes of signalling mechanism ..".

As reviewer 1 also highlighted the first sentence in the abstract, we have altered it to make it more clear.

Minor comments

1. Cumulative regulatory potential

From the abstract, it is not clear what the authors mean by 'Cumulative regulatory potential'. The readers have to come to page 9 lines 8 to 10 to understand the meaning.

As one of the main findings of the study, I think it should be explained in the abstract itself.

We have removed the somewhat confusing phrase from the abstract, and furthermore reduced the focus on the "regulatory" aspect of the modification throughout the text. We instead focus on a more direct definition: mediating a protein interaction.

2. Page 3 Line 5

The review Chen et al 2011 is majorly discussing arginine methylation, but not general PTM system. PMID 25692714 could be an appropriate review for providing background of the field to the general audience.

PMID 25692714 is a computational analysis, not a review. We will therefore also include a brief review into PTM systems for the general readership (Khoury, 2011)

3. Page 5 Line 5. "panel of 37 full length proteins" is misleading. If anything it is a rearrangement of RGG domains along the C terminal disordered region of SYNCRIP.

The RGG domains are not rearranged in the full length proteins referred to, they are mutated to lysines residues to preclude arginine methylation, we will therefore keep with the mutant full length protein description.

4. Page 6 line 7-1 n Figure 1B, sequence related features were analyzed rather than biophysical ones. Also, the title of figure caption describes the analysis as "biochemical" which is misleading.

Altered the text to reference sequence features rather than biophysical/biochemical.

5. page 6 line 8 - In order to provide a full context, the authors could mention the features of the other 70% of the arginine methylation sites. Also, this point could be carried to the Discussion section to explain whether the findings of the study are in any way informative in understanding the mechanisms of the remaining 70% of the Arginine methylation sites.

A very comprehensive bioinformatic analysis of arginine methylation has been undertaken previously (Sci Signalling, referenced in the text), therefore we specifically focused on the features of proteins harbouring clusters of arginine methylation.

6. Page 6 line 11- It is quite obvious that if there were more methyl Rs per 20 amino acid window, it is more likely that there would be more methyl Rs per RG motifs. The analysis from figure 1B could be presented as a supplementary information.

The figure does not show the ratio of methyl-R per RG motif, but the ratio of methyl-R in RG motifs. There is no reason that usage of the RG motif needs to increase as the density of methylation increase per se, any neighbouring amino acid could be present, yet from our analysis we can see that as density increases so does the ratio of sites in a RG motif.

7. Page 13 line 17- It would be combinations rather than permutations, as the order does not matter. They are all Arginines. Therefore the number of possibilities would be lower than what is estimated. If L is the length of the C terminal tail of SYNCRIP, combinations are $L!/(19! \times (L-19)!)$

We are referring to arginine to lysine mutations, therefore the position of the lysine mutation in the C terminal tail does matter (KGKGG_(...)RGRGG may behave differently to KGRGG_(...)RGKGG,) as such they are permutations not combinations. We have reworded the text to make this more clear.

8. The meaning of the sentences is quite unclear and would need improvement.

(1) page 8 lines 20 to 22 - Arginine is the identity of the amino acid

Altered the text to make more clear

(2) page 11 lines 5 to 7 - Confusing and repetitive.

We apologize but it is not clear to us which text is referred to here.

(3) page 16 line 5

We apologize but it is not clear to us which text is referred to here.

9. Typos

Page 5 line 17 - It's PhosphoSitePlus rather than phosphositeplus.

Altered in text

page 10 line 24 - By HNRP, the authors are probably referring to "heterogeneous nuclear ribonucleoproteins (hnRNP)".

Altered in text

10. Commas - Inserting commas between long sentences would improve the readability of few sentences. e.g. page 2 line 13

Ok

11. Page 8 line 20- Full form of the abbreviation Me-R should be declared.

Mentioned once in text by mistake, altered.

12. No section titles in the results.

We have added results section titles to make the text more clear.

13. Error in labeling the subplot.

(1) Figure 1C The key to the symbol 'D' is missing in.

Added to Figure

(2) Figure 1F Mentioned on page 8 line 3: location of the panel should be indicated. Rightmost in this case.

Added to Text

(3) Figure 1G Mentioned on page 8 line 20: there is no middle or right panel.

Altered in text, was a type as also noted by referee 1.

(4) Figure 2C It is mislabeled (as B) in the figure caption.

Altered Figure caption

(5) Figure 3B There is no middle panel.

Removed from text for clarity, we were referring to the middle blot of the three.

14. As the C terminal tail of SYNCRIP is approximately 100 amino acid long, the readers would benefit from visualizing the protein sequence in figures 1E, 2A and 4A.

In addition to the description in the text, for example: page 11 lines 8 to 9, a schematic with actual protein sequence would be more informative.

The protein sequence was presented in Supplementary Figure 1, but we have moved it to main Figure 2B to make it more clear what is the actual sequence we are mutating. This will also aid understand the description in the text.

15. Statistical testing- Wherever Mann-Whitney-Wilcoxon tests are used, in figure captions, it should be mentioned whether the test is one-sided or two-sided.

Altered in all Figure Legends.

16. Page 6 line 14- Wherever applicable, references to the methods sections should be included
ok

17. Page 20 line 4- Authors could have mentioned the overlap length. It could be an overlap of 5 residues leading to sampling bias. Ideally there would be an overlap of 1 residue.

The overlap analysis was undertaken as previously published in Woodsmith et al (Plos Comput Biol, 2013). The analysis is designed to identify the maximum density of modification in any given 20 amino acid window across a linear protein sequence. As such, windows require an overlap of 10 amino acids (as used here), to avoid false negatives. If the density results were added together, then we agree this overlap would cause a sampling bias. However, as it is just used to identify density maxima this is not an issue.

18. Structural analysis- Because it is a structural analysis, it would be useful to add a crystal structure, if available, in the supplementary information indicating location of methyl binding and which residues are mutated.

We have added the crystal structure and the mutated residues therein to Supplementary Figure 2.

19. Western blot

Figure 3A- Quantitative data should be represented with statistical tests applied over biological replicates.

The gel pictures (whole rather than cut-out/cropped) can be presented in supplementary information.

Quantifying western blots using two distinct antibodies (loading control and signal with distinct binding curves) is not recommended without quantifiable (i.e. pure protein) standards.

20. Image analysis

For Figure 3C -Authors could have shown an overlap of Hoechst and GFP filters. That would make the result clearer.

Done

Figure 3C and 3E- It would be better to substantiate the results related to colocalization experiments with appropriate statistical tests (such as in PMID: 24117417) applied.

The PMID 24117417 refers to particular interpretation of Pearson's correlation coefficient (PCC) and Manders' correlation coefficient (MCC) when applied to co-localisation studies. We have not used these tests in the analysis and argue that our original statistical analysis based on the experimental count data is robust enough to substantiate the conclusions drawn.

21. In case of 'GTEX dataset' and 'Gnomad data analysis', the websites and the date of accession should be mentioned in Methodology section.

We have referenced the precise file name we used for the analysis and where we downloaded the data from in the methods section.

22. Molecular docking.

In support of the major claim of the study that seemingly random positioning rather than contiguous arrangements of arginines allows better binding, the authors could try to simulate the binding using molecular docking. Otherwise, it is unclear why this would happen. As binding of proteins is the core subject of the study, a detailed structural analysis could provide valuable depth of information.

As far as we are aware, molecular docking approaches require structural information. As the methylation here is across a 100 amino acid unstructured region, statistical rigorous approaches in this direction are (with our expertise) currently unfeasible.

23. Page 2 line 6

The manuscript may benefit from extending on the lines of "tunable interaction surface". The aspect is largely untouched in the introduction and the discussion. There is evidence (PMID: 21868366) that alternate splicing generated isoforms that allow for the presence or absence of the motifs, providing the tunability. The manuscript would benefit from such a discussion of such events if they occur in the case of SYNCRIP.

We retrieved the 7 annotated isoforms of SYNCRIP annotated in NCBI. As can be seen from this sequence alignment, the number of arginines in the C terminus only varies by 3 in two of the isoforms, likely not representing a huge functional impact. We will therefore not add this discussion to the manuscript.

```

Iso1
QAAKNQMYDDYYYYGPPHMPPPTRGRGRGRGGYGYPPDYGYEDYYDYGYDYHNYRGG 480
Iso4
QAAKNQMYDDYYYYGPPHMPPPTRGRGRGRGGYGYPPDYGYEDYYDYGYDYHNYRGG 445
Iso7
QAAKNQMYDDYYYYGPPHMPPPTRGRGRGRGGYGYPPDYGYEDYYDYGYDYHNYRGG 328
Iso3
QAAKNQMYDDYYYYGPPHMPPPTRGRGRGRGGYGYPPDYGYEDYYDYGYDYHNYRGG 445
Iso5
QAAKNQMYDDYYYYGPPHMPPPTRGRGRGRGGYGYPPDYGYEDYYDYGYDYHNYRGG 480
Iso6
QAAKNQMYDDYYYYGPPHMPPPTRGRGRGRGGYGYPPDYGYEDYYDYGYDYHNYRGG 480
Iso2
QAAKNQMYDDYYYYGPPHMPPPTRGRGRGRGGYGYPPDYGYEDYYDYGYDYHNYRGG 382
*****

```

Iso1
 YEDPYYGYEDFQVGARGRGGRGARGAAPS RGRGAAPP RGRAGYSQRG GPGSARGV RGARG 540
 Iso4
 YEDPYYGYEDFQVGARGRGGRGARGAAPS RGRGAAPP RGRAGYSQRG GPGSARGV RGARG 505
 Iso7
 YEDPYYGYEDFQVGARGRGGRGARGAAPS RGRGAAPP RGRAGYSQRG GPGSARGV RGARG 388
 Iso3
 YEDPYYGYEDFQVGARGRGGRGARGAAPS RGRGAAPP RGRAGYSQRG GPGSARGV RGARG 505
 Iso5
 YEDPYYGYEDFQVGARGRGGRGARGAAPS RGRGAAPP RGRAGYSQRG GPGSARGV RGARG 540
 Iso6
 YEDPYYGYEDFQVGARGRGGRGARGAAPS RGRGAAPP RGRAGYSQRG GPGSARGV RGARG 540
 Iso2
 YEDPYYGYEDFQVGARGRGGRGARGAAPS RGRGAAPP RGRAGYSQRG GPGSARGV RGARG 442

Iso1
 GAQQQRGRGV RGARGGRG GNVGGKRKADGYNQPDSKRRQTNNQNWGSQP IAQQPLQGGDH 600
 Iso4
 GAQQQRGRGV RGARGGRG GNVGGKRKADGYNQPDSKRRQTNNQNWGSQP IAQQPLQGGDH 565
 Iso7
 GAQQQRGRGQGKGVEAGPDLLQ----- 410
 Iso3
 GAQQQRGRGQGKGVEAGPDLLQ----- 527
 Iso5
 GAQQQRGRG--GKGVEAGPDLLQ----- 561
 Iso6
 GAQQQRGRGQGKGVEAGPDLLQ----- 562
 Iso2
 GAQQQRGRGQGKGVEAGPDLLQ----- 464

Iso1
 SGNYGYKSENQEFYQDTFGQQWK 623
 Iso4
 SGNYGYKSENQEFYQDTFGQQWK 588
 Iso7
 ----- 410
 Iso3
 ----- 527
 Iso5
 ----- 561
 Iso6
 ----- 562
 Iso2
 ----- 464

24. Mutation rate and rate of evolution (or degree of polymorphism) at specific positions across the genome are different things. Mutations are the direct result of the process of error incorporation in the genome, not how much diversity there is in a population. Saying that some sites are more prone to mutations is therefore incorrect because it suggests that they have higher mutation rates. Most likely they do not but rather mutations at these sites are less likely to be removed by natural selection.

We agree with the referee's comments and have altered the text accordingly.

Reviewer #3:

Woodsmith et al., computationally analyse the publicly available data on arginine methylation (R-methyl) sites in human proteins and show that (i) R-methyl sites tend to cluster within disordered regions of proteins, and that the clustering is more pronounced in the presence of 'RG' motifs and (ii) such R-methyl clusters are enriched in RNA-binding proteins (RBPs). By integrating R-methyl data with human tissue-wide gene-expression data (GTEx), they show that genes that contain R-methyl clusters are low-abundant and have less expression variability, and interpret that the expression of these genes is tightly regulated. By analyzing human natural variation data (gnomAD), the authors show that R-methyl clusters, compared to isolated R-methyl sites, tend to accommodate more mutations and reason that an array of R-methyl sites, could compensate for the loss of individual modification sites. By experimentally investigating R-methyl cluster in the multiple 'RG' motif containing RBP SYNCRIP, they identify that (i) SMN1 and PHF1 bind to SYNCRIP in R-methylation and K-methylation dependent manner, (ii) PRMT1 methylates SYNCRIP R-sites, (iii) R-methylation is required for efficient recruitment of SYNCRIP to stress granules, though SMN1 recruitment could not be detected and (iv) modified and unmodified 'RG' motifs mediated interactions cumulatively. Furthermore, using mutagenesis studies, they show that SMN1 binding is affected by total arginine content while PRMT1 binding required contiguous arginine stretches and propose that there could be two different modes of binding within these regions. The manuscript is written well and the figures are elegantly presented.

While some of the results presented here are correlative, the study in general does not offer broad insights pertaining to R-methylation or deep insights regarding SYNCRIP interaction(s) and functions. We recommend reconsideration of this work upon major revisions as discussed below:

Major Comments

1. There is no clear demarcation between the main findings and the peripheral ones. This is a result of the general lack of interpretations and inadequate connection between different findings and a clear overall take-home message. It is not clear as to how the experimental findings build on computational investigations to advance our understanding. Furthermore, the validity of generalizing the interpretations on one protein to a proteome-scale is questionable (specifically claims in the fourth paragraph of Discussion section).

We have added the individual SYNCRIP data for methylation, disorder, expression analysis and genetic data from the GNOMAD dataset to highlight further how SYNCRIP is a representative example of the trends observed in the bioinformatic analysis (Figure 2A-C).

2. The connection between computational and the experimental parts is weak. A figure that shows the overlap between the proteins with high arginine methylation density with R-methyl transferase interaction could serve this purpose and render the selection of SYNCRIP more meaningful. Nevertheless, the authors need to connect experimental findings with computational results and provide concrete biological insights.

See above.

3. Computational investigations: (i) Effect size estimates for the observed differences need to be provided for each comparison.

Odds ratio effect sizes have been calculated and reported on figures 1A and B.

(ii) Are the trends in Fig 1A-1C statistically significant?

Yes, statistical significance has been added to all figures.

(iii) Enrichment of R-methyl clusters in disordered proteins, RBPs, tightly controlled proteins, and those with more mutations are all isolated observations. How are these observations interlinked and what is the connection with R-methyl clusters. In other words, there is no correlative or causal evidence to suggest that the observed trends are cause or consequence of clustered R-methyl sites.

We agree that disorder and clusters of methylation are tightly interlinked, in fact we provided experimental evidence to suggest a mechanistic reason for this relationship (Figure 4 and 5 and discussion). This model also provides plausible explanations for the trends observed in expression and genetic variation data (see one before last paragraph in discussion). Finally, clustered methylation targets are enriched in annotated RBP domains in comparison to highly methylated proteins that do not contain the clusters, suggesting a connection that is to do with the clusters and not with methylation per se. We have altered the discussion to make this more clear.

(iv) Tight regulation (Fig. 1F): The analyses show that proteins with R-methyl clusters are low-expression genes and have low-variability across different tissues. How this inform about tight regulation, as there is no analyses on synthesis or degradation rates? Even if it does, disordered proteins are known to be tightly regulation, R-methyl clusters are enriched in disordered proteins and hence one would expect this correlation. Why is this surprising or an important finding? As pointed earlier, the authors should dwell on the possible connection(s) of R-methyl clusters and the observed trends. In addition, they should consider comparing proteins with dispersed (non-clustered) R-methyl sites with those that contain clustered R-methyl sites.

The analysis does not show that R-methyl clusters are low expression genes, nor do we mention low expression in the text. Gene expression variation can be measured as a distribution across different samples, and we agree that it is impacted by both synthesis and degradation events. However, though we not treat these events individually we present an analysis on the end result of these processes, namely overall gene expression regulation over a large variety of tissues using a very powerful dataset. Finally, we did compare the clustered methyl-sites with non-clustered methyl sites (Figure 1F) to highlight the impact of clusters, over and above a background of non-clustered methylation.

Furthermore, the observations could be confounded by the differences in the total no. of tissues analyzed for proteins with R-methyl clusters and the controls.

Yes, however, this was controlled for in the initial analysis. As stated in the materials and methods we randomly sampled exactly the same amount of data from the total methyl-arginine dataset to compare with the clustered windows.

(v) It appears that the authors try to motivate their choice of SYNCRIP by looking for enrichment of only RBPs in proteins with R-methyl clusters. While this might be valid, a simple Gene Ontology enrichment analysis will provide information about other functional classes that are enriched in both in proteins with clustered and dispersed R-methyl sites.

A very comprehensive bioinformatic analysis including the suggested analysis of arginine methylation has been undertaken previously, and we referenced it in the original text as follows: "Arginine methylated proteins have

been shown to be involved in multiple facets of RNA processing and binding, for example proteins containing RRM and RH RNA binding domains are preferentially modified (Larsen et al. 2016).”

(vi) How was the 20 amino acid window determined? How will the results change at different window sizes?

In our previous work we have tested varying the size of the windows used in PTM scanning analysis for other PTMs (20 & 50 amino acids, Woodsmith Plos Comp Biol 2013), showing no impact on the conclusions drawn.

4. Experimental investigations: The outcome of the experimental investigation is convoluted. (i) Only 15 of the 19 R-sites investigated, are known to be methylated. PRMT1 binding does not require methylated arginines (Fig. 3B). Though SMN1 requires R-methyl sites for binding, it is not clear which non-mutated R-sites are methylated. Therefore, it is not clear if the observed effects are because of clustered arginines or clustered 'RG' motifs or clustered 'R-methyl sites'.

We have clearly demonstrated the methylation dependency of the interaction (see above reviewer comments), therefore ruling out clustered arginines being responsible for the interaction. As for all motif based substrate recognition effects, the glycine in the RG motif plays a role in the interaction. Like a proline in CDK phospho sites, the non-modified residue in the motif is necessary but not sufficient for modification (many many peptide studies address the G in the RG motif). We therefore can conclude it is clustered methylated arginines present in RG motifs responsible for the SMN1 interaction.

(ii) What is the rationale behind selecting the sites that were mutated in 6Rs and 14Rs?

The 6R and 14R mutants were selected as they showed equivalent expression to wild type SYNCRIP in the stable cell lines we generated, while having either a low or intermediate number of arginines in comparison to the wild type.

(iii) Clustering the heatmaps in Fig. 4A based on the binding signal (SMN1 and PRMT1) might be more helpful to interpret the results and follow the description. Details of the controls used in Fig. 4B-4C needs to be provided.

We have added the details of the controls to the materials and methods section, and found that ordering by binding signal to be less helpful than the original. We feel that the order better reflects the systematic approach taken and thus eases the interpretation.

(iv) No statistical significance assessment or any statistical metric is provided for Fig. 4D-4G. Therefore, it becomes a bit difficult to assess the importance of these findings. Furthermore, in Fig.4D-4E, the bin sizes are variable between the different R-mutants. Please consider providing continuous scatter plots to avoid such variability in bin sizes.

Statistics added to figures 4D to E. As can be seen for the following figure, the trends are the same for the continuous data, however to be able to conduct sensible statistics on the data as suggested by the reviewer, we need to keep the data in the larger bins present in Figure 4.

(v) The R-mutant bins are again misleading here. For instance, the 10-13 bin essentially includes all the data points ranging from 10-12 considered in the 9-12 bin. Would generating distinct bins or plotting this data on continuous scales, lead to the same interpretations?

Our original analysis is more systematic than that suggested by the referee, and allows the statistics requested in point (iii) to be undertaken. Distinct bins are present in the diagram with the barplots at 9-12/13-16 and 12-15/16-19 modifications present and showing the same trend. By systematically increasing the methylation by one in each window, we both show the impact across all methylation levels and that no specific cut-offs are responsible for the patterns observed. The most important conclusion is drawn from comparisons between continuous and non-continuous of the same bin.

Minor Comments

5. The focal point of this study appears to be 'binding potential of R-clusters' (well presented in Fig. 5). Therefore, the title and the repeated discussion about 'cumulative regulatory potential' is misleading and often confusing. What regulation are the authors referring to?

We have altered the text to be more specific and refer to the interaction.

6. Usage of semantics: Low structural complexity and low-complexity sequences have been used synonymously. The authors have not computed any sequence-complexity measures (such as those computed by SEG) and should consider using disordered regions instead of low-structural complexity or low complexity. Similarly, phrases such as regulatory capacity, regulatory potential and regulatory interactions, need to be explicitly defined or avoided. As such none of the experiments done investigate the regulatory nature of the R-methyl sites. Please consider using non-stress instead of basal for sub-cellular localization experiments.

We have altered the text to always refer to low structural complexity to avoid any confusion with low complexity. We have also removed regulatory potential and regulatory interactions from the text for clarity, as well as using "non-stress" in the text for the sub-cellular localisation experiments.

7. The Results section should be divided into sub-sections with the overall finding/conclusion of each sub-section provided as title, for better comprehension of the findings.

Sub heading provided in results section.

8. Residue numbers, such as R409 and R411, used to represent 'R' mutants (in 6Rs and 14Rs) are difficult to follow as there is no reference provided in the manuscript. Please consider providing the residue numbers in Supplementary Figure 1C. Alternatively please use 3-letter code (e.g. Arg-409, etc.).

We have provided the protein sequence (and residue positions in new Figure 2B to aid better understanding of which arginines are mutated.

9. The last paragraph in the introduction section about di-amino acid repeats is abrupt, as there is no clear logical flow from R-methylation, R-content to di-amino acid repeats.

We removed the phrase from the text to more specifically refer to RG motifs.

10. Ratio in the y-axis label of Fig. 1A, 1B and 1D should be proportion or fraction. No. of datapoints in each distribution needs to be mentioned. In Fig 4D-4G, should the y-axis label be Log2 binding signal?

Figures altered as suggested.

11. The distribution profile of disorder for each amino acid across the length of SYNCRIP should be provided as a supplementary figure.

Added into new Figure 2A.

September 10, 2018

RE: Life Science Alliance Manuscript #LSA-2018-00178-TR

Prof. Ulrich Stelzl
University of Graz
Universitätsplatz 1
Graz A-8010 Graz
Austria

Dear Dr. Stelzl,

Thank you for submitting your revised manuscript entitled "Interaction modulation through arrays of clustered methyl-arginine protein modifications". As you will see, the reviewer who re-evaluated your manuscript appreciates the introduced changes and thinks that your manuscript is suitable for publication in Life Science Alliance. We would be happy to publish your paper in Life Science Alliance pending final revisions necessary to meet our formatting guidelines.

- please provide all figure files as individual files without legend and include the figure legends in the main manuscript docx file instead
- please list 10 authors et al in your reference list
- please make sure that all corresponding authors provide an ORCID iD.

A. FINAL FILES:

-- High-resolution figure, supplementary figure and video files uploaded as individual files: See our detailed guidelines for preparing your production-ready images, <http://life-science-alliance.org/authorguide>

-- Summary blurb (enter in submission system): A short text summarizing in a single sentence the study (max. 200 characters including spaces). This text is used in conjunction with the titles of papers, hence should be informative and complementary to the title. It should describe the context and significance of the findings for a general readership; it should be written in the present tense and refer to the work in the third person. Author names should not

be mentioned.

B. MANUSCRIPT ORGANIZATION AND FORMATTING:

Full guidelines are available on our Instructions for Authors page, <http://life-science-alliance.org/authorguide>

Sincerely,

Andrea Leibfried, PhD
Executive Editor
Life Science Alliance
Meyershofstr. 1
69117 Heidelberg, Germany
t +49 6221 8891 502
e a.leibfried@life-science-alliance.org
www.life-science-alliance.org

Reviewer #3 (Comments to the Authors (Required)):

This is a revised manuscript resubmitted after review by the group of Stelzl. The authors have carefully considered all the comments of the three referees and have addressed all comments

as constructively as possible without the additional experiments. They have also modified the manuscript accordingly and discuss potential limitations of the interpretations supported by the data in a balanced manner. In short, I support publication of the revised version of the ms in LSA as presented and look forward to seeing this interesting work in print.

September 12, 2018

RE: Life Science Alliance Manuscript #LSA-2018-00178-TRR

Prof. Ulrich Stelzl
University of Graz
Universitätsplatz 1
Graz A-8010 Graz
Austria

Dear Dr. Stelzl,

Thank you for submitting your Research Article entitled "Interaction modulation through arrays of clustered methyl-arginine protein modifications". It is a pleasure to let you know that your manuscript is now accepted for publication in Life Science Alliance. Congratulations on this interesting work.

The final published version of your manuscript will be deposited by us to PubMed Central (PMC) as soon as we are allowed to do so, the application for PMC indexing has been filed. You may be eligible to also deposit your Life Science Alliance article in PMC or PMC Europe yourself, which will then allow others to find out about your work by Pubmed searches right away. Such author-initiated deposition is possible/mandated for work funded by eg NIH, HHMI, ERC, MRC, Cancer Research UK, Telethon, EMBL.

Please also see:

<https://www.ncbi.nlm.nih.gov/pmc/about/authorms/>

<https://europepmc.org/Help#howsubsmanu>

*****IMPORTANT:** If you will be unreachable at any time, please provide us with the email address of an alternate author. Failure to respond to routine queries may lead to unavoidable delays in publication.*******

DISTRIBUTION OF MATERIALS:

Again, congratulations on a very nice paper. I hope you found the review process to be constructive and are pleased with how the manuscript was handled editorially. We look forward to future exciting submissions from your lab.

Sincerely,
